# Quantitative imaging datasets of surface micro to mesoplankton communities and microplastic across the Pacific and North Atlantic Ocean from the Tara Pacific Expedition

Zoé Mériguet[1], Guillaume Bourdin[2], Nathaniel Kristan[2], Laetitia Jalabert[3], Olivier Bun[1], Marc Picheral[3], Louis Caray–Counil[1], Juliette Maury[1], Maria-Luiza Pedrotti[1], Amanda Elineau[3], David A. Paz-Garcia[4], Lee Karp-Boss[2], Gaby Gorsky[1], Fabien Lombard[1] and Tara Pacific Consortium Coordinators team[+]

[1] Laboratoire d'Océanographie de Villefranche-sur-Mer, Sorbonne Université, CNRS, France
[2] School of Marine Sciences, University of Maine, Orono, Maine 04401, USA
[3] Sorbonne Université, CNRS, Institut de la Mer de Villefranche, IMEV, 06230 Villefranche-sur-Mer, France
[4] Laboratorio de Genética para la Conservación, Centro de Investigaciones Biológicas del Noroeste, Baja California Sur 23096, México.
[+] A full list of authors appears at the end of the paper.

*Correspondence to*: Zoé Mériguet (zoe.meriguet@-imev-mer.fr) and Fabien Lombard (fabien.lombard@imev-mer.fr)

**Abstract.** This paper presents the quantitative imaging datasets collected during the Tara Pacific Expedition (2016-2018) on the schooner Tara. The datasets cover a wide range of plankton sizes, from micro-phytoplankton > 20 μm to meso-zooplankton of a few cm, as well as non-living particles such as plastic and detrital particles. It consists of surface samples collected across the North Atlantic and the North and South Pacific Ocean from open ocean stations (a total of 357 samples) and from stations located in coastal waters, lagoons or reefs of 32 Pacific islands (a total of 228 samples). As this expedition involved long distances and long sailing times, we designed two sampling systems to collect plankton while sailing at speeds up to 9 knots. To sample microplankton, surface water was pumped onboard using a customised pumping system and filtered through a 20 μm mesh size plankton net (here after Deck-Net (DN). A High Speed Net (HSN; 330 μm mesh size) was developed to sample the mesoplankton. In addition, a Manta net (330 μm) was also used when possible, to collect mesoplankton and plastics simultaneously. We could not deploy these nets in reef and lagoon stations of islands. Instead, two Bongo nets (20 μm) attached to an underwater scooter were used to sample microplankton. In addition to describing and presenting the datasets, the complementary aim of this paper is to investigate and quantify the potential sampling biases associated with these two high speed sampling systems and the different net types, in order to improve further ecological interpretations. Regarding the imaging techniques, microplankton (20-200 μm) from the DN and Bongo nets was imaged directly on-board Tara using the FlowCam (Fluid imaging, Inc.) while the mesoplankton (> 200 μm) from the HSN and Manta nets was analyzed in the laboratory with the ZooScan system, back on land. Organisms and other particles were taxonomically and morphologically classified using the web application EcoTaxa automatic sorting tools, followed by taxonomic expert validation or correction. For micro-plankton smaller than 45 μm, a subsample of 30% of the annotations was 100% visually validated by experts. More than 300 different taxonomic and morphological groups were identified. The datasets include the metadata with the raw data from which morphological traits such as size (ESD) and biovolume have been calculated for each particle, as well as a number of quantitative descriptors of the surface plankton communities. These include abundance, biovolumes, Shannon diversity index and normalised biovolume size spectra, allowing the study of their structures (e.g. taxonomic, functional, size structure, trophic structure, etc.) according to a wide range of environmental parameters at the basin scale.

## 1. Introduction

Zooplankton serve as an important conduit for the transfer of energy from primary producers to higher trophic
levels (Ikeda, 1985). In this key position in the food webs, they also play an important ecological and
biogeochemical role (Turner, 2015; Steinberg and Landry, 2017), with associated ecosystem services. In
particular, they are essential to Pacific fisheries management, as they influence fish productivity and ecosystem
dynamics (Balachandran and Peter, 1987; Chuanbo Guo et al., 2019; Hays, 2005). The datasets we present here,
cover a wide diversity of surface plankton, ranging from 20 μm to few cm, at the scale of the Pacific Ocean. The
vastness and unique characteristics of the Pacific Ocean make it a particularly interesting study area. From
nutrient-rich upwelling or islands zones to oligotrophic gyres, the diverse oceanic processes of the Pacific Ocean
present a wide range of environmental conditions that significantly influence plankton communities, making it a
key region for plankton research (Chavez et al., 2011; Longhurst, 2007). However, sampling efforts of
zooplankton in the Pacific Ocean largely focused on the temperate North Pacific, eastern and western boundary
currents in the North Pacific, leaving vast areas under-sampled (Drago et al., 2022). This gap is particularly evident
in the NOAA zooplankton dataset (https://www.st.nmfs.noaa.gov/copepod/atlas), where the under-sampling is
particularly true for the central subtropical and tropical Pacific where fisheries are important resources for the
thousands of pacific islands. We present a map (Fig. 1) overlaying updated zooplankton databases with samples
from the Tara Pacific expedition, illustrating how these new data address sampling gaps. Global mapping of
zooplankton in the Pacific is hindered by the highly expansive operational ship time face to this vast ocean. The
use of high-speed sampling, such as the Continuous Plankton Recorder (CPR, by Hardy in 1926), the LHPR
(Longhurst et al., 1966), the Gulf III OCEAN Sampler (Gehringer, 1958), the Gulf V plankton sampler (Sameoto
et al., 2000), as well as newer low-tech designs (CSN in Von Ammon et al., 2020; Coryphaena in Mériguet et al.,
2022), including the one employed in our datasets, provides valuable opportunities to expand sampling coverage
and frequency and thus address this undersampling. In the hope of increasing similar cruising speed zooplankton
sampling efforts, we discuss the benefits, challenges and limitations of this high-speed sampling approach based
on the lessons learned from obtaining these datasets.

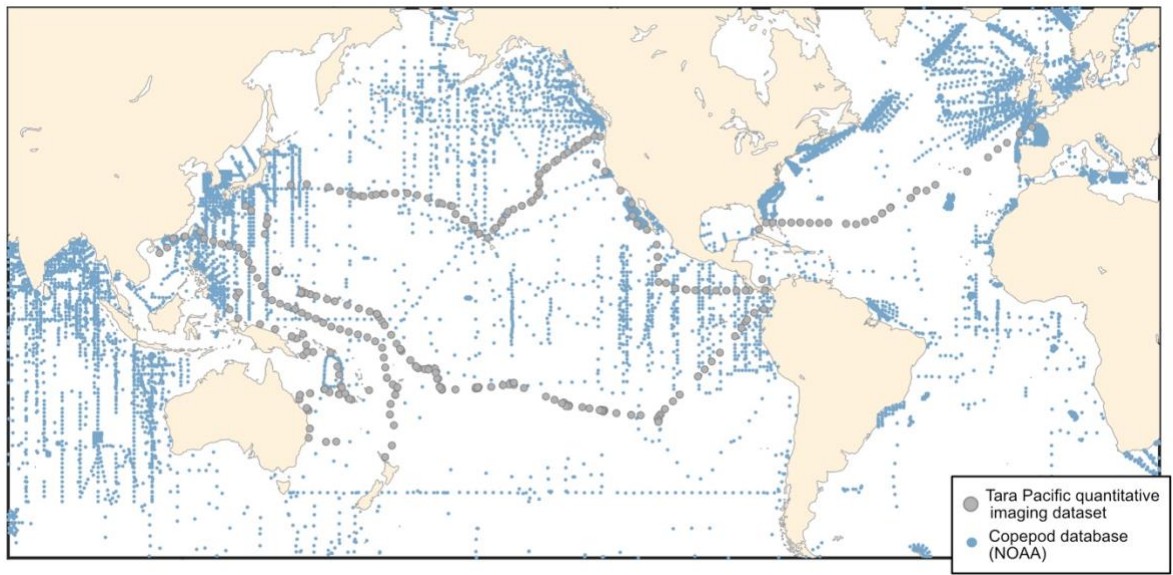

**Figure 1. Spatial distribution of zooplankton observations from the COPEPOD database**
**(https://www.st.nmfs.noaa.gov/copepod/; all groups) is represented by blue points. Plankton imaging data (> 20 μm)**
**from the Tara Pacific expedition are shown in grey.**

The aim of this paper is therefore to present and discuss this open-access quantitative plankton imaging datasets
sampled during the Tara Pacific Expedition (2016-2018), conducted in the Pacific Ocean. In general, the effects
of different environmental forcings on plankton are often focusing on one size range of plankton, or on a particular
taxonomic or functional type to the exclusion of others. It is often difficult to reconcile different methods of

analysis (taxonomic, biogeochemical, genomic) to provide a coherent view of the plankton as a whole. In this respect, quantitative imaging is complementary to other methods to study plankton community composition (e.g. HPLC, flow cytometry, genomics) because it simultaneously provides quantitative measures of abundance, morphology and biovolume (as a proxy for biomass) for different taxonomic groups of plankton organisms (Lombard et al., 2019). The datasets represent a diversity of surface plankton analysed with the use of two quantitative imaging instruments: 1. the FlowCam (Sieracki et al., 1998), which images microplankton from 20 to 200 µm, and 2. the ZooScan (Gorsky et al., 2010), which images meso-zooplankton (>200 µm). The dataset also includes the plastics imaged by the ZooScan. Overall, it encompass a total of 2 356 231 images, including both surface micro and mesoplankton, as well as non-living particles such as plastics, making a significant contribution to improving the availability of plankton data.

These datasets are of great value because of the relative rarity of sampling surface planktonic communities at the oceanic scale. Potential limitations of the data presented here are discussed below. To ensure adequate spatial coverage while considering navigation constraints, we designed two new sampling systems to collect surface micro- and mesoplankton while sailing at a maximum speed of 9 knots. The 'Dolphin' sampler was designed to pump seawater into a 20 µm net on board, the Deck Net (DN), while the 'High Speed Net' (HSN) was designed and towed to collect surface plankton larger than 300 µm in size (see Gorsky et al., 2019 for details). In addition to these high-speed sampling devices, but with less extensive spatio-temporal coverage, a Manta net (330 µm) was also used whenever cruising speed made it possible (*i.e.* < 4 knots), to collect surface mesoplankton and plastics. Two Bongo nets (20 µm), towed by an underwater scooter, were also used by scuba divers around islands, reefs, and lagoons. Thus, a complementary objective of this paper is study and quantify the potential sampling biases of the different methods used during this expedition, in order to maximize the quality of the data offered to the scientific community and promote similar high speed zooplankton sampling efforts which strongly enhance the spatial coverage of samples. Another characteristic of these datasets is the daytime sampling of surface (0-1 meter) plankton communities. This offers the possibility of geographic intercomparisons and interdisciplinary studies related to the ocean's surface layer, enabling direct comparisons with other surface measurements, such as satellite and atmospheric data. However, this raises questions about the quantitative nature of the sampling itself, particularly regarding the representativeness of the datasets. While these datasets provide quantitative accuracy by offering all the necessary information to consistently calculate estimates of the sample contents, we must warn that the data may not fully be 'quantitatively representative' of the broader ecosystem. Although the sampling objective is the surface layer, daytime sampling alone cannot document the nocturnal intrusion of migrating zooplankton and micronekton to the surface. It is worth mentioning that night sampling was also operated on zooplankton alone (see Fig 10 in Gorsky et al 2021) but therefore does not reconcile in space and time with day sampling and was therefore not analyzed in priority.

## 2. Methods

### 2.1 Sampling

We present a collection of FlowCam and ZooScan images acquired during the Tara Pacific expedition (2016-2018; Gorsky et al. 2019, Lombard et al. 2023). All samples and protocol names in this article follow Lombard et al. (2023) in order to help the user match the samples and associated data presented here with other samples from the expedition. Sampling was carried out generally at the daily frequency, every ~150-200 nautical miles, during daytime, resulting in a total of 249 sampling events labelled [oa001] to [oa249] (Fig. 2). The first 28 sampling events occurred during the trans-Atlantic crossing as the ship sailed from France to the Pacific. At the end of the expedition, the schooner Tara acquired quantitative imaging samples at stations [oa232] to [oa249] across the North Atlantic. Data are published on the SEANOE platform to allow for future updates and completion of datasets. The plankton sampling covers a large latitudinal range (temperate, subtropical, and tropical) as well as a diversity of environments associated with different oceanic regimes (equatorial upwelling, coastal upwelling, eastern boundary current, subtropical gyres, and other provinces). We collected over 357 samples in the open ocean and 228 samples close to the reef or in the lagoon. A selection of 32 coral reef islands systems (labelled

[i01] to [i32]) in the tropical and subtropical Pacific Ocean were targeted for coral reef holobiont studies (Planes
et al., 2019), including surface plankton sampling analysed by quantitative imaging. A summary of geological,
topological and human population characteristics of the different islands targeted (name, size, elevation, human
population, etc.) can be found in Lombard et al. (2023). Any sampling event that was conducted within the
Exclusive Economic Zone (EEZ) of an island (defined as the area that stretches 200 nautical miles or 370 km out
of the coastline of an island in question) was considered as an island station and annotated with the island label
[i##_oa###]. All other sampling events were considered open ocean stations (high seas, 132 open ocean stations)
and were annotated [i00_oa###].

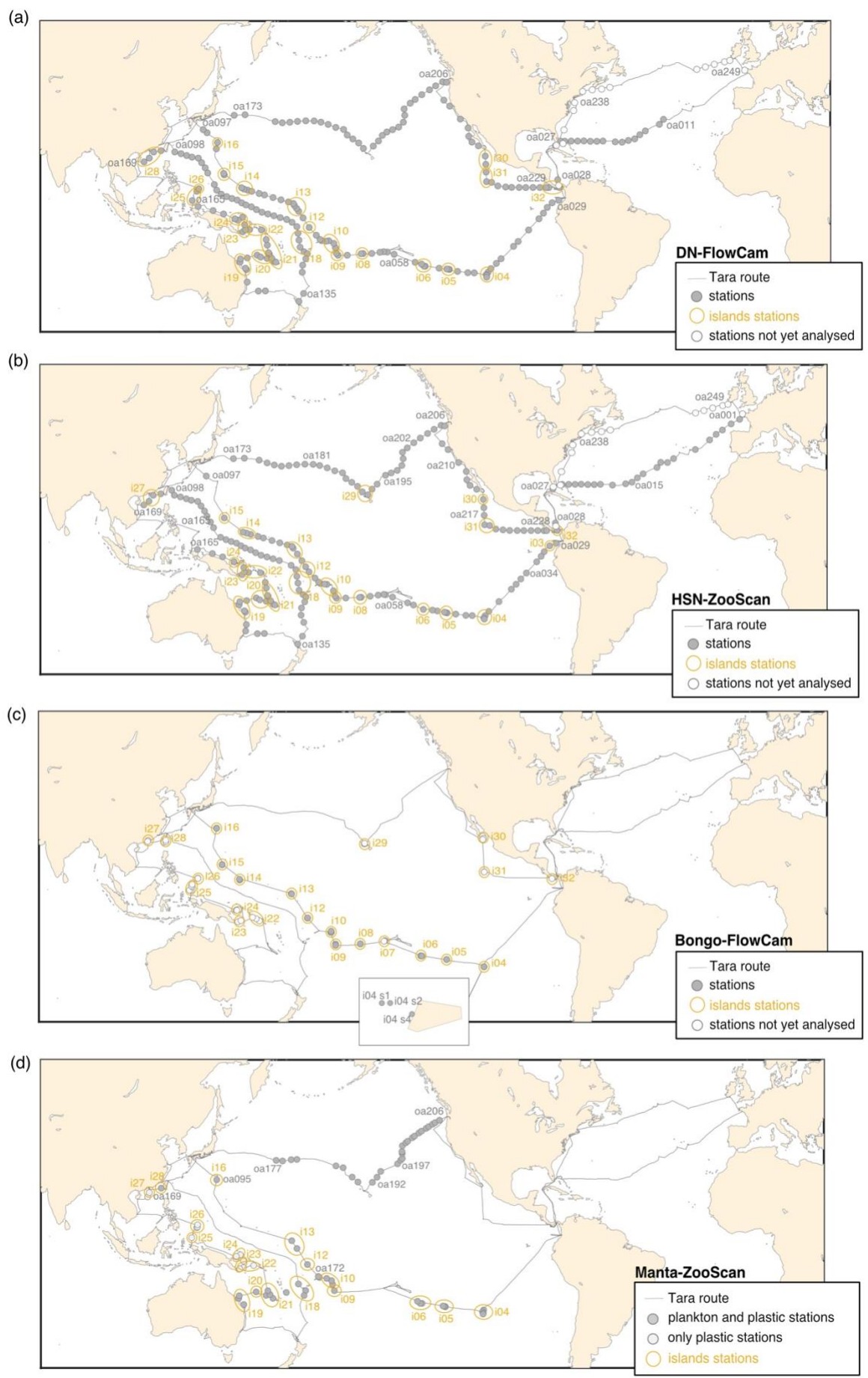

**Figure 2. Tara Pacific expedition (2016–2018) sampling map for the 4 different datasets. Continuous sampling: (a) DN (Deck-Net) – FlowCam (b) HSN (High-Speed-Net) – ZooScan. More discrete sampling, focus around islands: (c) Bongo Net – FlowCam and (d) Manta - ZooScan (plankton and plastic samples). Island stations, station within 200 nautical miles of an island, are represented inside a yellow circle. The 'not yet analysed' stations in the figure legend mean that the samples have not yet been scanned for the ZooScan dataset and have not been taxonomically validated for the FlowCam dataset.**

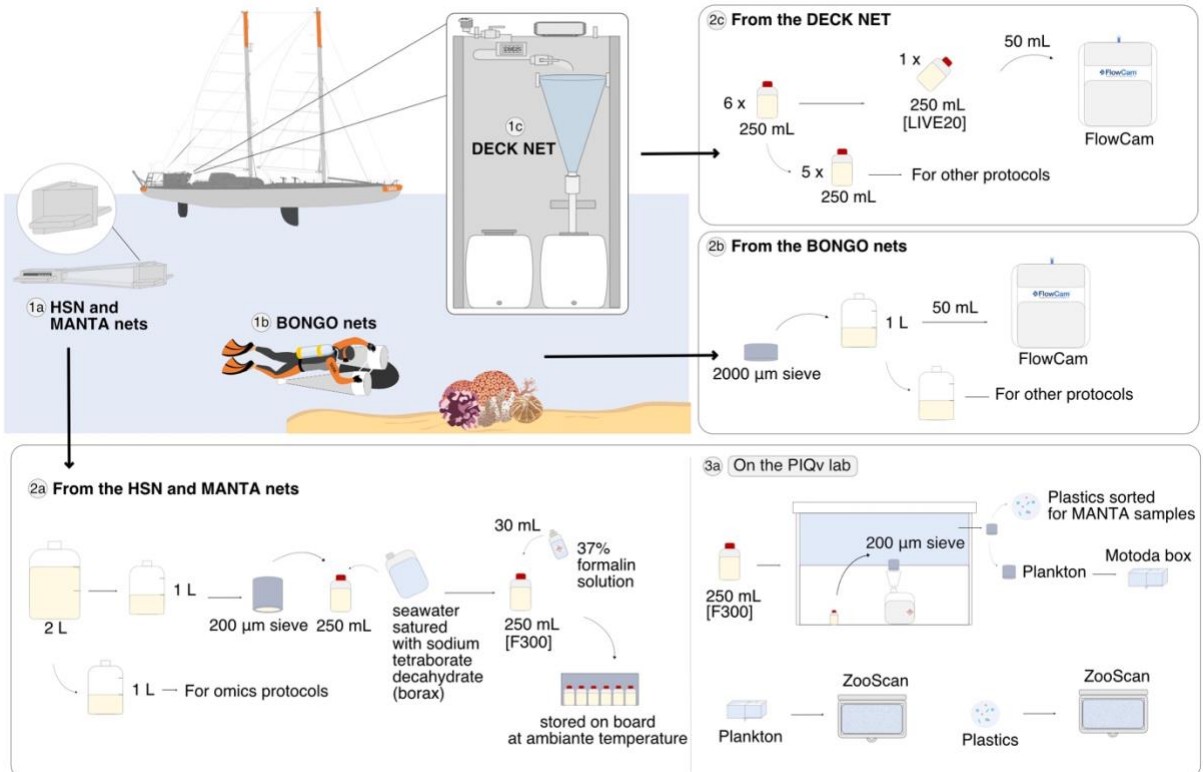

**Figure 3. Schematic overview of the sampling events and protocols used during the Tara Pacific expedition for quantitative imaging. The top left panel corresponds to the sampling events with the deployed plankton nets: (1a) the 330 μm High Speed Net (HSN) and the 333 μm Manta net, (1b) the 20 μm Bongo nets attached to the underwater scooter and (1c) the 20 μm Deck Net (DN) on the deck of the Tara. Samples from DN (2c) and Bongo (2b) were imaged live with the FlowCam (20-200 μm) and samples from HSN and Manta (2a) were imaged with the ZooScan (> 300 μm). For the ZooScan analysis, samples were fixed using formaldehyde and stored on board and analysed on the Imaging Quantitative Platform (PIQv) in the laboratory in Villefranche-sur-Mer, the protocols in this platform are detailed in the section: "On the PIQv lab" (3a). Somes drawings were taken from Lombard et al. 2023 modified (credit N. Le Bescot).**

### 2.1.1 Deck-Net sampling

Surface water samples were collected using a custom-built water pumping system named "Dolphin". It consists of a stainless-steel pyramidal frame with a front aperture of 0.04 m wide and 0.40 m high, deployed from the starboard side of the ship (see pictures in Gorsky et al., 2019). The Dolphin was used underway while sailing and was connected to a peristaltic pump (max flow rate = 3 m$^3$ h$^{-1}$) mounted on the deck of the schooner Tara. The system was equipped with a flowmeter to record flow rates. The pumped water was filtered through a 20 μm net (Deck-Net) that was mounted on the wall of the wet lab (Fig. 3; 1c and pictures in Gorsky et al., 2019). Before entering the Deck-Net, the pumped water passes through a 2000 μm mesh filter. Deck-Net pumping lasted 1 to 2 hours, depending on plankton concentration. Samples were divided into subsamples, which included one subsample for quantitative micro-plankton imaging analysis on live samples (LIVE20; Fig. 3; 2c) and the remaining for specific protocols detailed in Lombard et al. (2023). Further information on the Dolphin system, the Deck-Net, and various protocols based on this sampling can be found in Gorsky et al. (2019) and Lombard et al. (2023).

### 2.1.2 Bongo nets sampling

Plankton larger than 20 μm were sampled at ~2 m below the sea surface using two small diameter Bongo plankton nets with 20 μm mesh size and an opening area of 0.071 m$^2$. These nets were towed by divers using underwater scooters (Fig. 3; 1b) and towed for about 15 min at maximum speed ($0.69 \pm 0.04$ m s$^{-1}$). Each net was equipped with a flowmeter rated to provide accurate measurements at speeds above 0.3 m.s$^{-1}$, but, the relatively low maximum speed of the underwater scooter was insufficient to allow seawater to flow through the 20 μm mesh fast enough to trigger the rotation of the flowmeter. Therefore, volume was estimated from the tow speed and tow duration using the following Eq. (1):

$$\text{Bongo volume} \ = \ 0.071 \times \text{tow speed} \ \times \ \text{tow duration} \tag{1}$$

### 2.1.3 HSN and Manta nets sampling

Simultaneously with the deployment of the Dolphin to collect microplankton, the High Speed Net (HSN) was towed to sample the mesoplankton. The HSN was equipped with a 330 μm mesh and designed to be deployed while sailing up to 9 knots (average speed deployment: 6.7 knots). The HSN features the same mouth opening as the Dolphin system, consisting of a stainless-steel pyramidal frame with a front aperture measuring 0.40 x 0.04 m (see zoom on the HSN mouth system on Fig. 3). The base opening of this pyramidal structure measures 0.34 x 0.34 m. This net was deployed from the starboard side and towed at a distance of 50–60 m behind the ship (to avoid it being in the wake of the ship), for a period of 60–90 min (depending on plankton density). In addition to the HSN, Manta net was also deployed in some locations (Fig. 2). The Manta net have rectangular frame of $0.16 \times 0.60$ m mouth opening with a 4 m long net with 333 μm mesh size, and was used at a maximum speed of 3 knots, for an average of 30-40 minutes.

Flowmeters were mounted at half of the opening height above the bottom of the opening on both HSN and Manta nets to ensure it was well submerged during deployment while measuring the filtered volume. Theoretical volumes were calculated taking into account a 3/4 mouth opening of the HSN and Manta nets, $0.3 \times 0.04$ and $0.6 \times 0.12$ m, respectively (see Eq. (3), (4) and (5)). As these nets are surface nets, the water collected actually passed through ~3/4 of the opening height (see photos of deployments in Gorsky et al., 2019). To calculate volumes from the flowmeter for the HSN, we considered an opening of $0.34 \times 0.34$ m, corresponding to the dimensions of the pyramid base opening where the flowmeter was positioned inside the HSN (Eq. (2)). We compared the volume estimated from the flowmeter readings with theoretical estimation using the towing distances. We computed the towing distances using the minute binned latitude and longitude recorded with the Tara's GPS along each deployment. We calculated the distance between the start-end latitude and start-end longitude for each minute, to calculate the distance per minute covered by the boat. We then summed these 'per-minute' distances over the duration of the deployment to obtain a calculated distance that is as close as possible to the true towing distance and accounts for potential modification of the boat's heading during deployments. The equations for calculating the filtered volumes are therefore as follows. The 0.3 factor in the flowmeter volume equation corresponds to the impeller pitch, as recommended by Hydrobios, to convert the number of revolutions into towing distance.

$$\text{HSN flowmeter volume} \ = \ \text{flowmeter end} \ - \ \text{flowmeter start} \ \times \ 0.3 \ \times \ (\text{HSN mouth opening area}) \tag{2}$$

$$\text{HSN theoretical volume} \ = \ \text{tow distance} \ \times \ (\text{HSN mouth opening area}) \tag{3}$$

$$\text{Manta flowmeter volume} \ = \ \text{flowmeter end} \ - \ \text{flowmeter start} \ \times \ 0.3 \ \times \ (\text{Manta mouth opening area}) \tag{4}$$

$$\text{Manta theoretical volume} \ = \ \text{tow distance} \ \times \ (\text{Manta mouth opening area}) \tag{5}$$

Simplified Metadata in csv provides both flowmeters and theoretical volumes for HSN and Manta net, enabling the user to select the filtered volume for the calculation of quantitative descriptors. A discussion of the biases associated with each estimate is given in section 3.2. The filtered volumes uploaded as metadata in EcoTaxa (EcoTaxa export table in tsv, see part 2.5) and used to compute quantitative descriptors (see part 2.5) are the theoretical volumes calculated from the distance (see the results of technical validation part 3.2.1).

Once recovered, samples collected both by the HSN net and the Manta net followed the same procedure (Fig. 3;
2a). The sample was divided into two 1 L fractions (details in Gorsky et al., 2019). One fraction was concentrated
on a 200 μm sieve and resuspended in a 250 mL double-sealed bottle using filtered seawater saturated with sodium
tetraborate decahydrate (borax), fixed with 30 mL of 37% formalin solution and stored at room temperature for
taxonomic and morphological analysis by imaging methods in the laboratory (samples named [F300]). The other
fraction was used for omic analysis.

**2.2 Acquisition and treatment of plankton imaging data**

Sample labels were annotated by different users at different times during the expedition and are therefore not
homogeneous. In order to avoid confusion or misunderstanding of the labelling of the samples, an additional
column has been created in the csv Simplified Metadata (column "Homogenous sample names") with
homogeneous names for all datasets.

**2.2.1 FlowCam analysis**

Samples from the Deck-Net (250 mL) and Bongo net (50mL) were imaged live directly on board using a FlowCam
Benchtop B2 series (Fluid Imaging Technologies; Sieracki et al., 1998) equipped with a ×4 objective and a 300
229 μm deep glass flow cell to examine the micro-plankton samples (size range 20-200 μm: Fig. 3; 2c). Each sample
was first passed through a 200 μm sieve to remove large objects that could clog the FlowCam imaging cell.
Samples were then diluted or concentrated to achieve optimum object flow. The auto-image mode was used to
image the particles in the focal plane at a constant flow rate.

**2.2.2 ZooScan analysis**

The ZooScan imaging instrument (Gorsky et al. 2010) was used to study the mesoplankton. Samples collected
from the HSN and Manta nets ([F300]) were imaged at the Quantitative Imaging Platform (PIQv) of the Institut
de la Mer de Villefranche (Fig. 3; 3a). In addition, preserved zooplankton samples are stored in the Collection
Center for Plankton of Villefranche (CCPv). The formaldehyde solution was replaced by filtered seawater during
the analysis.

*Plankton samples analysis from HSN and Manta nets on the ZooScan*

Before scanning on the ZooScan, plankton samples were divided using a Motoda splitter (Motoda, 1959) to obtain
a concentration of approximately between 1000 and 2500 objects per subsample and scanned with the ZooScan.
This sampling strategy correctly accounted for the many small organisms as well as the large ones that might be
under-sampled when subsampling with the Motoda box. This limit ([1000- 2500] objects) was defined by the
PIQv platform to avoid the overlap of planktonic organisms, while retaining enough organisms to give a reliable
quantitative measurement of the sample. After each scan, a quality control was systematically carried out
concerning i) the quality of the scanned image and ii) the number of objects imaged, to ensure that that the number
of objects is within the limits given above. The quality control tool for imaging data is accessible on the PIQv
website: https://sites.google.com/view/piqv/. After treatment in the ZooScan, all samples were re-concentrated on
a 200 μm sieve and resuspended in a 250 mL double-sealed bottle using filtered seawater saturated with borax,
fixed with 30 mL of 37% formalin solution and returned to the CCPv.

The borax (sodium tetraborate decahydrate) used as a buffer may form crystals grains, forming white crystals. If
the borax solution was not filtered sufficiently, crystals would end up in the plankton samples, be digitised and
counted as objects. Thus, if Borax was not filtered sufficiently, white crystals may represent a large proportion of
the objects within the 1000-2500 limit and thus bias the quantitative measurement of the plankton. We identified
24 samples containing borax crystals during the analysis. Therefore, prior to scanning, these samples were
thoroughly rinsed with filtered seawater through a 300 μm mesh sieve to remove a maximum of borax crystals

from the sample. A 200 µm mesh sieve was placed below the 300 µm sieve in order to conserve the initial sample
in the collection (CCPv). Analysis on the ZooScan was performed from the 300 µm sieve.

*Plastic sampling from Manta net*

Samples from the Manta nets were gently transferred to a Petri dish. Plastic-like particles were manually separated
from other components such as wood, zooplankton, and organic tissues (Fig. 3; 3a). Entangled pieces of plastic
were picked up manually from zooplankton and aggregated under a stereoscopic dissecting microscope, using
forceps. The visual criteria used to classify a microfiber as synthetic were the absence of cellular structures and
scales on the surface, a curved shape with a uniform surface, a uniform thickness along the entire length of the
filament, spots, and strong strands (Barrows et al., 2018; Hidalgo-Ruz et al., 2018). Each sample was examined
twice to ensure the detection of most of the plastic particles. Isolated plastic particles were then imaged with
Zooscan. To minimise the plastic contamination of the samples, a quality control approach was undertaken
following the protocol described by Pedrotti et al. (2022).

**2.3 Images processing**

For FlowCam and ZooScan, the full methodology used can be found in their respective manuals
(https://sites.google.com/view/piqv/piqvmanuals/instruments-manuals; for the ZooScan the protocol is also
available on zenodo by Jalabert, 2022). Images generated by FlowCam and ZooScan were processed using the
ZooProcess software in ImageJ (Gorsky et al. 2010) which extracts segmented objects as vignettes. During this
process, each vignette was associated with a set of 46 morphometric measurements for object characterization,
including grey levels, fractal dimension, shape and size, which were imported into the EcoTaxa web application
(Picheral et al. 2017) for taxonomic classification. For ZooScan, the ZooProcess software includes a tool that
enables the digital separation of potentially touching or overlapping objects in the original image. If two objects
(possibly two plankton organisms) are touching, they will be considered as a single vignette and assigned a single
label, which could therefore biais estimates of abundance and size, as described in Vandromme et al. (2012).
Objects that were still touching after the application of the ZooProcess automatic tool were identified and
separated using the ZooProcess manual separation tool to improve the quality of the subsequent taxonomic
annotation, counts and size structure analysis of the zooplankton. For each ZooScan dataset, this quality control
step was systematically performed during taxonomic annotation.

**2.4 Taxonomic identification**

Using image recognition algorithms on EcoTaxa, predicted taxonomic categories were validated or corrected by
trained taxonomists. For the majority, the taxonomic classification effort was possible up to the genus and only in
rare cases up to the species. A number of organisms could not be reliably taxonomically identified due to a lack
of identification criteria and were therefore grouped into temporary categories (t00x) following similar
morphological criteria. Nine different trained taxonomists from the PIQv platform annotated FlowCam and
ZooScan vignettes on these datasets. Annotations of FlowCam and ZooScan vignettes from the different nets were
also done by different taxonomists but the list and the global criteria to identify a group were common. To reduce
operator bias between taxonomists and to ensure taxonomic consistency, a final stage of homogenisation was
carried out by two taxonomists after all vignettes had been validated. At the time of publication of these datasets,
copepod genera had not been homogenised for ZooScan, but homogenisation will be pursued in the future and the
published SEANOE dataset will be updated accordingly. Overall, these datasets are published on the SEANOE
flexible platform that allows updates and corrections, so that taxonomic annotations can be improved over time.
All vignettes with taxonomic annotations are visible on the open access project in EcoTaxa (section 4).

**2.5 Case of FlowCam taxonomic identification for objects smaller than 45 µm**

The Tara Pacific settings for the FlowCam live analysis generates many more images than the ZooScan. For
example, for station oa140, the ZooScan counts 1 435 images compared to 42 915 images for the FlowCam. Given

that taxonomists annotated images on an image-by-image basis, the validation or correction of the automatic
classification on these numerous FlowCam images would require a much higher investment of time than for the
ZooScan samples. In addition, the resolution of the FlowCam images of the smallest organisms does not allow us
to classify them properly and at a sufficient precision. Therefore, we validated only 30% of the total images
smaller than 500 pixels (equivalent to ~45 μm in ESD), randomly picked, assuming that this 30% random
subsample leaves a statistical count that is sufficiently representative of the population. Prior to this choice, a
series of tests were conducted to assess the impact of different fraction of image validation at varying object size
thresholds. Samples were randomly selected and 100% of the images were taxonomically validated. Subsequently,
a series of simulations (three times for the four samples, random sampling each time) were conducted to assess
the impact of varying size thresholds (i.e. from 200 to 600 pixels, equivalent to 18 to 55 μm, with a step of 50
pixels) on the proportion of total images to be annotated (fractions from 5% to 50%, with increments of 5%). We
compared the results of these simulations by using the relative Root Mean Square Error (RMSE). The RMSE
values were divided by the total number of 100% validated values and multiplied by 100 to express the cumulative
error as a percentage. Results are shown in Fig. 4 and illustrate the cumulative error across the absolute abundance
values. For our chosen threshold of 500 pixels and subsets at 30% (highlighted in bold on the Fig. 4), we observed
induced errors of 0.02%. In Figure 4d, we present the absolute abundance and taxonomic group composition of
plankton from the four samples that were 100% taxonomically annotated, alongside the same four samples that
were only 30% (< 500 pixels) annotated. These samples show highly comparable results in both absolute
abundance and taxonomic composition (data not shown). We carried out the same analysis as described in Figure
4 for the total size spectrum, slope of the NBSS, and for the taxonomic composition (relative abundance). They
showed an induced error of 20% and 12%, respectively. This supplementary analysis can be found in appendix C.
The software ZooProcess 8.27, available on the PIQv website, now includes the capability for subsampling on
Flowcam data.

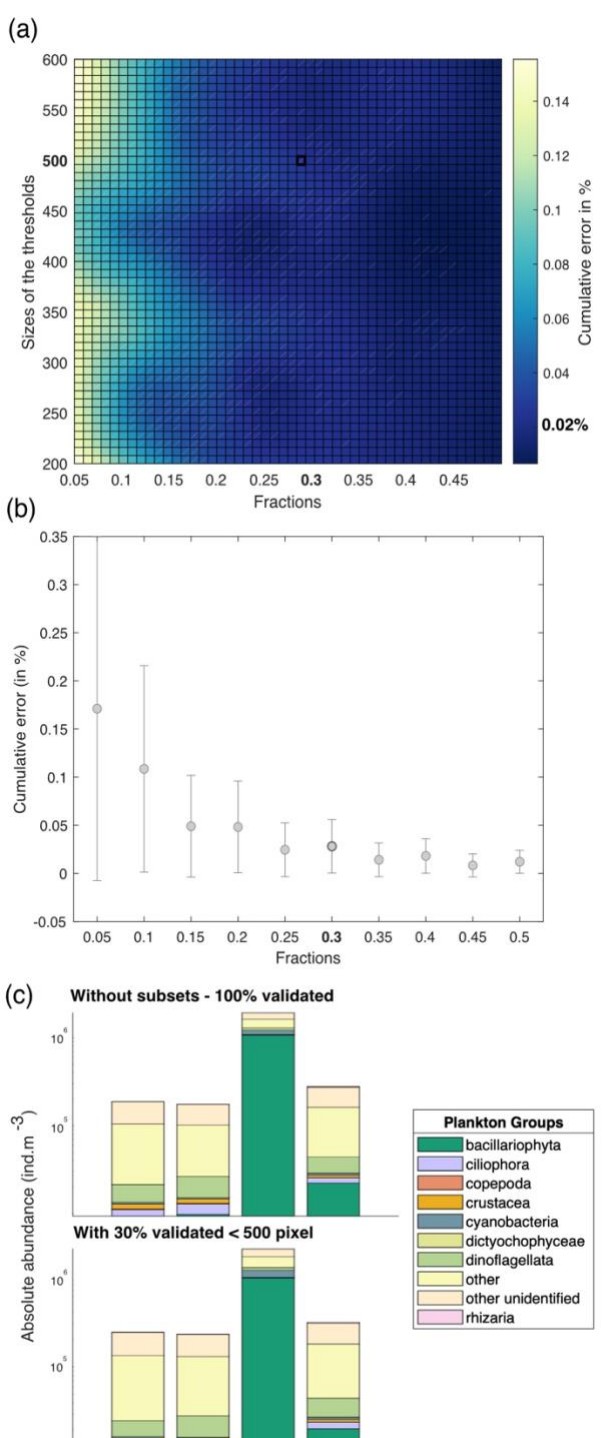

**Figure 4. (a) Estimated cumulative error associated with partial validation of particles below a size cut-off threshold**
**ranging from 200 to 600 pixels and validated fractions ranging from 5% to 50%. Errors are computed as the percentage**
**Root Mean Squared Error (RMSE) between fully validated samples and partially validated samples in three different**
**metrics for cumulative error in absolute abundance. RMSE values represent the outcomes of simulations, each**
**conducted three times for the four samples, with random sampling. (b) Cumulative error according to the Fractions**
**chosen. The threshold is fixed at 500 pixels. (c) Comparison between the absolute abundance (ind.m⁻³) and plankton**
**group composition for samples taxonomically annotated at 100% and for the same samples annotated at 30% below**
**the threshold of 500 pixels, equivalent to 45 μm.**

**2.5 Datasets**

**2.5.1 Plankton images on EcoTaxa and the associated tsv.**

The datasets include 4 datasets of microplankton imaged by the FlowCam and sampled by the Deck-Net and the
Bongo Nets, and mesoplankton imaged by the ZooScan sampled by the HSN and the Manta. All of the sorted
images of plankton, plastic and particles are visible on the open-access projects on the EcoTaxa web application.
The *.tsv files exported from the EcoTaxa platform are provided. Readme tables for FlowCam and ZooScan *.tsv
are also provided to facilitate their use.

**2.5.2 Quantitative descriptors to study the micro- and meso-plankton community**

For each dataset, we designed a table combining the metadata and data from which we have calculated quantitative
descriptors of planktonic communities: abundance (ind/m$^3$), biovolume (mm$^3$/m$^3$; proxy of biomass) and Shannon
diversity Index. Abundance (ind/m$^3$) and biovolume (mm$^3$/m$^3$) were calculated taking into account the volume of
water filtered by the plankton samplers (see formula in Table 1). Biovolumes (in mm$^3$/m$^3$) were computed using
area, riddled area, and ellipsoidal measurement of each object, and are available in the *.csv table (following
Vandromme et al., 2012; formula in Table 1). For analysis shown here, major and minor axes of the best ellipsoidal
approximation were used to estimate the biovolume of each object, following the recommendations of
Vandromme et al. (2012). Size was expressed as equivalent spherical diameter (ESD, μm, see formula Table 1).
Diversity was calculated using the Shannon index (H: see formula Table 2). It is important to note that Shannon's
diversity index is dependent on the number of taxonomic categories, as defined by Shannon and Weaver (1949),
it assumes that individuals are randomly sampled from an independent large population and that all species are
represented in the sample. However, in the majority of cases, taxonomic classification was possible up to genus
level using quantitative imaging methods. This must be taken into account in these Shannon diversity indices,
which therefore differ from more commonly used taxonomic categories. The individual biovolumes of the
organisms were arranged in Normalised Biomass Size Spectra (NBSS), as described by Platt & Denman (1978),
along a harmonic range of biovolumes such that the minimum and maximum biovolumes of each class are linked
by: $B_{vmax}= 20.25 \ B_{vmin}$. The NBSS was obtained by dividing the total biovolume of each size class by its biovolume
interval ($B_{vrange}=B_{vmax}-B_{vmin}$). The NBSS was representative of the number of organisms (abundance within a
factor) per size class. This can provide insight into ecosystem structure and function through the 'size spectrum'
approach, which generalises Elton's pyramid of numbers (Elton, 1927, Sheldon, 1972, Trebilco et al., 2013). The
NBSS size spectra of each sample (in abundance/μm) is provided in a separated zip files (.csv). Plankton
abundance and biovolume were calculated for each taxonomic annotation and for different levels of grouping:
living or nonliving, plankton groups and trophic association. The full list of these groups linked to all EcoTaxa
taxonomic annotations is given in the Table A1 to A4 (appendix A) of the taxonomic list and groups in each
dataset.

| Descriptors | | Formulas for FlowCam | Formulas for ZooScan |
|---|---|---|---|
| **Abundance** (ind/m$^3$): Number of individus in the sampling/ m$^3$ | | (object_annotation_category x sample_conc_vol_ml) / (acq_fluid_volume_imaged x sample_initial_col_vol_m3) | (object_annotation_category x acq_sub_part) / sample_tot_vol |
| **Biovolume** (m m$^3$/ m$^3$): Volume biomass of individus in the | Plain biovolume | $(4/3 \ x \ \prod x \ ( \ \sqrt{(object\_area)} / \prod) )^3 \ x$ sample_conc_vol_ml) / (acq_fluid_volume_imaged x sample_initial_col_vol_m3) | $(( 4/3 \ x \ \prod x \ (\sqrt{(object\_area)} / \prod )^3) \ x \ acq\_sub\_part) /$ sample_tot_vol |

| sampling/ m³ | Riddled biovolume | $(4/3 \times \prod \times (\sqrt{(object\_area\_exc\ (mm2) / \prod})^3 \times sample\_conc\_vol\_ml) / (acq\_fluid\_volume\_imaged \times sample\_initial\_col\_vol\_m3)$ | $((4/3 \times \prod \times (\sqrt{(\sqrt{(object\_area\_exc / \prod})})) 3) \times acq\_sub\_part) / sample\_tot\_vol$ |
|---|---|---|---|
| | Ellipsoid biovolume | $(4/3 \times \prod \times [(object\_major/2) \times (object\_minor/2) \times (object\_minor/2)] \times sample\_conc\_vol\_ml) / (acq\_fluid\_volume\_imaged \times sample\_comment\_or\_volume)$ | $((4/3 \times \prod \times [(object\_major\ (mm)/2) \times (object\_minor\ (mm)/2) \times (object\_minor\ (mm)/2)]) \times acq\_sub\_part) / sample\_tot\_vol$ |
| **Diversity** Shannon Indice (H) | | $-\sum (abundance\ relative\ (\%) / 100) * log(abundance\ relative\ (\%) / 100)$ | |
| **Equivalent Spherical Diameter** (ESD, μm) | | $2 \times \sqrt{(object\_area \times process\_pixel^2 / \prod)}$ | |

**Data description**

object_area : surface area of the object [pixel²]
object_area_exc : surface area of the object excluding holes (object_area*(1-(object_%area/100)) [pixel²]
object_minor :  length of secondary axis of the best fitting ellipse for the object [pixel]
object_major : length of the primary axis of the best fitting ellipse for the object [pixel]
process_pixel : dimension of the side of a pixel in the scanned image [mm]

**Data description for FlowCam**
See Export EcoTaxa FlowCam read me.csv

object_annotation_category : taxon display_name in Ecotaxa
sample_conc_vol_ml : concentrated or diluted water volume (from sample_comment_or_volume) [mL]
acq_fluid_volume_imaged : flowcam total images volume [mL]
sample_initial_col_vol_m3 : initial collected volume, (if nets : sum of the nets) [mL]

**Data description for ZooScan**
See Export EcoTaxa ZooScan read me.csv

object_annotation_category : taxon display_name in Ecotaxa
acq_sub_part : subsampling division factor of the sieved fraction of the sample
sample_tot_vol : total filtered volume by the sampling gear [m3]

**Table 1. Formulas used to calculate quantitative variables in datasets. The variable names correspond to the real names**
**of the variables in the exports (tsv files) and are described in the table.**

**3. Technical validation and discussion**

**3.1 Limitations of Bongo net micro-plankton sampling for quantitative estimations**

Both the Bongo nets and the Deck Net consisted of a 20 µm mesh to collect surface micro-plankton throughout
the expedition. A key difference between these two nets lies in their deployment locations, which correspond to

distinct environments: Bongo nets were deployed near islands, reefs, or within lagoons, while the Deck Net was
deployed in the open ocean. These environments are characterized by differing chlorophyll a concentration, with
a clear increase observed near islands and within lagoons, as highlighted in Bourdin et al. (2024). As such, we
expected higher plankton concentrations in the reef and lagoon areas, and consequently, in the Bongo net samples.
However, the majority of Bongo net samples showed lower concentrations than nearby open ocean samples from
the Deck Net, as evidenced by the NBSS size spectra (Fig. 5a).

This discrepancy raises concerns about our reliability of the volume-filtered estimates, whether based on
flowmeters or theoretical calculations, which are critical for consistent quantitative plankton sampling. Regarding
the flowmeters, as mentioned in the methods section, Bongo nets were equipped with flowmeters rated for speeds
above 0.3 m·s$^{-1}$. However, the relatively low towing speed of the underwater scooter was insufficient to generate
enough water flow through the 20 µm mesh to rotate the flowmeters reliably. For the theoretical volume, the
deployment time of the Bongo nets by divers was highly uncertain. The uncertainty surrounding the theoretical
volume stemmed from inconsistent deployment times recorded by the divers and methodological biases associated
with using an underwater scooter, which made the filtered volume estimates unreliable. Moreover, the suspended
particle concentrations were very variable for different sampling sites which complicated the correct prediction
of the towing time required to obtain reasonable concentrate in the net and avoid clogging.

Overall, the lack of correlation of total chlorophyll a and total phytoplankton biovolume from FlowCam, as shown
in figure 5b, indicates that the Bongo net sampling was not quantitative. The chlorophyll a (chla) values obtained
from the HPLC measurements do not represent the same size classes of phytoplankton as those observed with the
FlowCam, but we were interested in whether or not there were likely to be similar trends in phytoplankton biomass
changes measured for the same station (Fig. 5b). The correlation between chlorophyll a and total phytoplankton
biovolume of the Bongo being lower than for the Deck-Net samples. This suggests that phytoplankton biovolume
was underestimated relative to chlorophyll a in the Bongo samples. Given the methodological limitations of the
Bongo net filtration volume estimation, our most plausible hypothesis is an overestimation of the theoretical
volume likely due to clogging. Therefore, as a conclusion, it is highly recommended to use Bongo net samples for
qualitative analysis only.

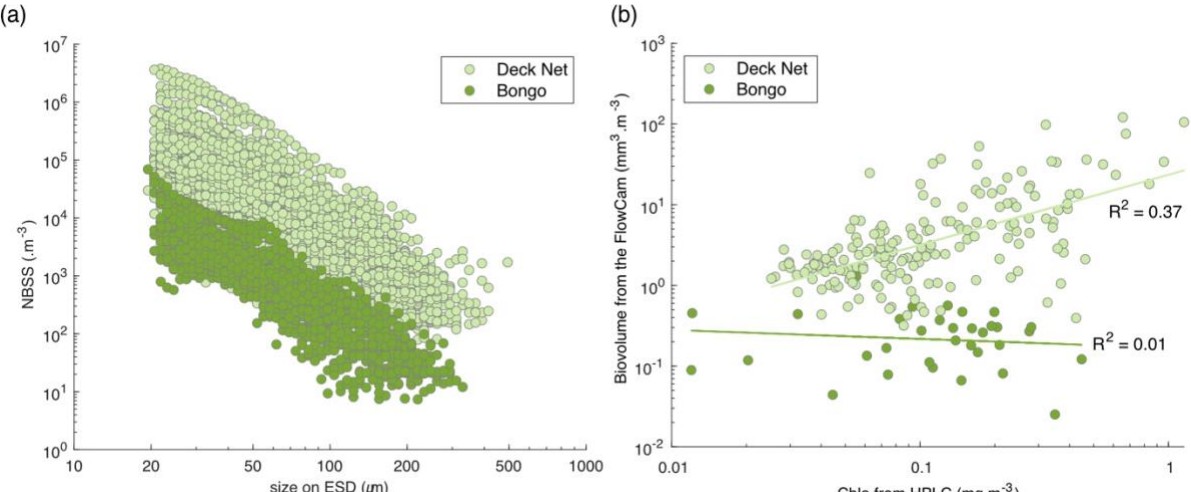

**Figure 5. (a) Comparison of Normalised Biovolume Size Spectra (NBSS; in log-log) of the live plankton between the**
**Bongo nets (34 samples) and the Deck Net (207 samples). (b) Phytoplankton biovolume (mm³.m⁻³) estimated from the**
**FlowCam samples, which were collected using the Bongo nets and the Deck Net, according to the Chla values obtained**
**from the HPLC measurements at the same station. The selection of phytoplankton organisms was made possible by**
**taxonomic validation of FlowCam images from these two nets.**

**3.2 Benefits and limitations of high-speed deployment**

During the Tara Pacific open ocean transects, we decided to take on the challenge of collecting plankton samples while sailing at speeds of up to 9 knots. This high-speed sampling provides valuable opportunities to expand and optimise the coverage of our sampling with a daily frequency. Initially, the Tara Pacific expedition was designed to focus on coral reefs (Planes et al., 2019). The addition of high-speed sampling allowed for the opportunistic use of transit periods, covering a significant spatial area of the expedition. As a result, one of the most valuable aspects of the Tara Pacific plankton samples is the daily collection of samples approximately every 150 to 200 nautical miles, covering a wide range of oceanic structures across the Pacific basin. However, it is important to note that, given the patchy spatial distribution of plankton (Robinson et al., 2021), this sampling scale is somehow discrete rather than continuous. This designed sampling is also valuable as we aimed for 'end-to-end' sampling of surface waters (Gorsky et al., 2019) with the micro to macroplankton fractions presented in this article. However, the constraint of surface sampling and of deploying and retrieving the instruments at cruising speed forced us to develop new robust, relatively small and user-friendly devices adapted for the Tara schooner. The combined deployment of the Dolphin system and the High-Speed Net (HSN) designed to this purpose and present in this article, represents, to our knowledge, the first system enabling discrete sampling of the entire surface planktonic ecosystem with deployment and retrieval at cruising speeds < 9 knots.

The development of the high-speed plankton samplers began in the early 20th century with the well-known Continuous Plankton Recorder (CPR), developed by Alister Hardy in 1926, which is designed to be towed under the surface over long distances at speeds up to 25 knots. Following the CPR, other high-speed net systems emerged, including the Longhurst-Hardy Plankton Recorder (LHPR: Longhurst et al., 1966), Gulf III OCEAN Sampler (Gehringer, 1958), and Gulf V plankton sampler (Sameoto et al., 2000) as well as newer low-tech designs (CSN in Von Ammon et al., 2020; Coryphaena in Mériguet et al., 2022). All high-speed zooplankton samplers face the challenge of maintaining filtration efficiency at higher towing speeds. Thus, higher speeds require a larger relative filtration area to optimises filtration efficiency while minimising excessive pressure on the net and mitigating the pressure wave that pushes organisms away from the net (Harris et al., 2000; Keen, 2013; Skjoldal et al., 2013). A critical design principle is therefore to obtain a sufficiently high ratio of mesh filtering area to net opening area (Smith et al., 1968b; Skjoldal et al., 2013). To achieve this, high-speed zooplankton samplers often employ a small initial opening area that widens internally (e.g. CPR has an 1.27 cm$^2$ entrance aperture expanding to 5cm x 10cm; the use of conic noses on the Gulf-V and LHPR). This design trade-off essential for pressure reduction, comes at a cost. The small surface area of the mouth opening means a smaller volume filtered, reducing the probability of collecting less abundant, larger organisms (Skjoldal et al., 2013). The avoidance of active swimming zooplankton, net opening area size dependent, is also described as the bias affecting the catch of mesoplankton by Harris et al., 2000. This may be discussed, as increasing tow speed may improve the capture efficiency of zooplankton capable of active avoidance (Skjolad et al. 2013). Therefore, high-speed sampling methods have the advantages of increasing sampling coverage and frequency, but they also introduce bias due to the pressure generated by high speeds, resulting in even greater undersampling compared to traditional nets (Harris et al., 2000; Cook and Hays, 2001).

**3.2.1 Impact on filtered volumes estimation**

One of the primary challenges in quantitative plankton sampling is the estimation of the filtered volume. Because the immersion depth of surface nets changes constantly with waves, wind and boat movement, it is difficult to accurately calculate the volume of water being filtered (reviewed in Pasquier et al., 2022). Results obtained by different studies show that a surface sampling with a difference in immersion depth of a few centimeters can lead to a large difference in the sampled volume (Pasquier et al., 2022). Overall, the impact of high-speed deployment on filtered volume remains largely unexplored in the literature with the exception of Jonas et al (2004). They tested the relationship between CPR filtered volumes estimated by a flowmeter or by theory, and their relationship to CPR deployment speed. Their findings revealed overestimations by the flowmeter compared to theoretical values. This raises concerns about the effectiveness of flowmeters in measuring volumes during high-speed

deployments. We therefore investigated whether our high-speed surface sampling approach had an effect on
filtered volume measurements.
For the Deck Net, the water intake was identical in design and mouth opening to HSN but a flowmeter was
integrated into the water circuit downstream of the pump as well as two de-bubblers (pictures Fig. 6 in Gorsky et
al., 2019). This allowed for reliable estimation of water volumes that were pumped into the Deck-net based on
flowmeter recordings (Gorsky et al., 2019). Both HSN and Manta nets were equipped with mechanical flowmeters
mounted in the inlet frame, while the towed distance, time and speed were recorded on board to also estimate the
theoretical volume filtered. While the HSN was towed between 3.9 and 9 knots, the Manta net was towed at lower
speed, between 1.2 knots and maximum speed of 3.6 knots (Fig. 6).

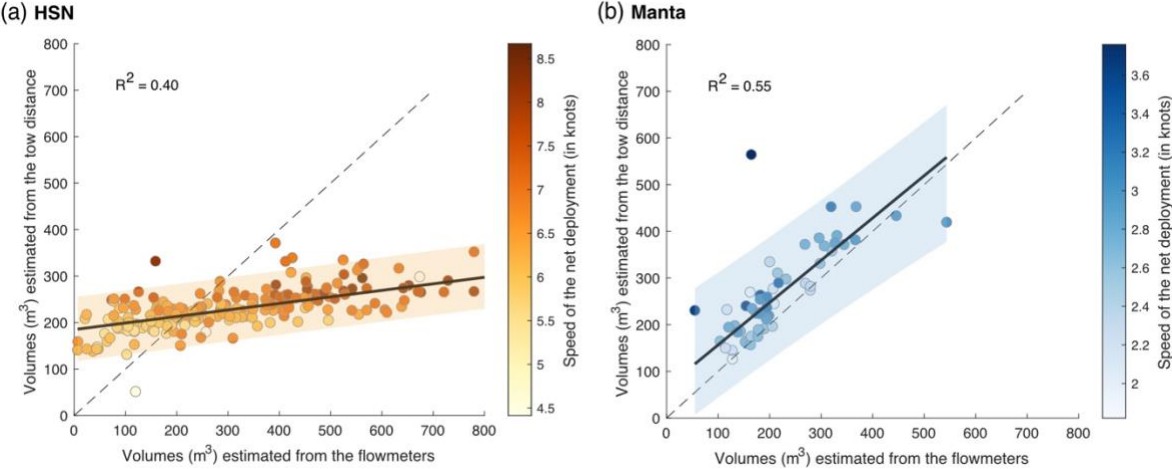

**Figure 6. (a) and (b) Linear regression between volumes filtered estimated from the tow distance (theoric volumes; m3)**
**and estimated from the flowmeters respectively for the HSN and Manta. The range of 95% confidence intervals is**
**represented in orange for the HSN and in blue for the Manta. The 1:1 dotted line represents the linear regression**
**obtained if both volumes were similar. The colour of the dots represents the deployment speed of the net in knots.**
Figure 6 shows a clear discrepancies in the slope of the estimated volumes between the HSN and the Manta,
meaning that the theoretical and flowmeter filtered volumes of the Manta are closer to each other than for the
HSN. Manta theoretical volumes tend to be higher and thus potentially overestimated compared to flowmeter
measurements (Fig. 6b), but the difference remains largely small compared to the HSN. For this one, flowmeter
estimation methods provide volumes in the same order of magnitude as the theoretical volume for HSN, yet exhibit
considerable differences between stations (mean difference between flowmeter and theoretical volumes per station
= 90.5, standard deviation = 172.6; Fig. 6a). Linear regression analysis between this volume differences per station
(flowmeters - theoretical volume) and speed deployment showed a significant relationship with a slope coefficient
of 91.168 (standard deviation = 11.86, t-test = 7.69 and p-value < 0.001), indicating that higher speeds are
associated with greater differences. Consistently with the results of Jonas et al (2004) described before, the high-
speed deployment is thus associated with the overestimation of the flowmeters volumes compared to theoretical
ones (Fig. 6a). These results indicate that the use of the flowmeters is not appropriate in high-speed conditions.
The pressure increase caused by the high speed generates turbulence and could affect the flowmeter rotation and
explain the overestimation of the filtered volume for the high-speed that we found. Globally, the turbulence
generated could explain the malfunction of flowmeters which are designed and calibrated by the manufactures to
accurately measure flow speed in a laminar flow. This result is highlighted by Skjoldal et al. (2019), who assume
the use of flowmeters being complex because of their position in relation to the cross-sectional flow field or
functioning in a turbulent system.

In addition to the speed, we tested the HSN's immersion depth varied when the sea state was high. The HSN was designed to sample the surface ocean, at the air-seawater interface, thus the upper part of its mouth opening was rarely completely submerged during the deployment (see images Fig. 4 in Gorsky et al. 2019). The relationships between wind strength (as a proxy for sea state) recorded by Tara's navigation instruments and the two estimates of HSN sampling volumes showed no correlation ($R^2 = 0.00$ for flowmeter volumes and for theoretical volumes; data not shown). While the flowmeter does not provide accurate flow measurements under turbulent conditions, it appears that the sea state does not affect its volume estimates.

Therefore, we recommended using the theoretical volume for the HSN. The towing distance used is relative to ground, not to the seawater, therefore there is a potential bias in the theoretical volume estimation due to the non-consideration of the surface current speed. This bias is likely negligible for the majority of our samples located in the subtropical gyres, mostly characterised by relatively low geostrophic currents (Tara Pacific data available Bourdin et al. 2022 in 'at current_speed_copernicus').

### 3.2.2 Quantitative comparison between HSN and Manta

The Manta net was designed to study neuston and floating particles, such as microplastics. Thus, it is the most commonly used net for studying surface plankton and widely recognised as a reference system for investigating surface ocean (Eriksen et al., 2018; Karlsson et al., 2020; Pasquier et al, 2022). Both HSN and Manta nets were deployed at the same stations when approaching islands and in the Great Pacific Garbage Patch. The Manta net was deployed in closer proximity to islands than the HSN net. Given that the HSN net was towed for a duration of 60–90 minutes, while the Manta net was towed for approximately 30–40 minutes, the decision was taken to sample with the Manta net in the immediate vicinity of the island, in order to capture the variability associated with the island mass effect.

We conducted a comparison of the Normalized Biovolume Size Spectra (NBSS; Fig. 7a) obtained from the two nets. The analysis follows the analysis presented in Lombard et al. (2023), incorporating data from 31 additional samples collected by the HSN. The NBSS of both nets was of the same order of magnitude, with Manta biovolumes appearing higher in each NBSS size class (Fig. 7a), suggesting an underestimation by the HSN. Considering the principle that, when represented on a logarithmic scale (as in Fig. 7c), the intercept of NBSS spectra reflects the total abundance of organisms in the studied ecosystem (Platt & Denman, 1978), and assuming the same water masses were sampled, we compared the NBSS intercepts, which support the underestimation by the HSN, as higher intercepts were observed for the Manta (with the NBSS intercept of HSN showing 0.2 compared to 0.8 for the Manta). This difference was expected due to the undersampling at high speed compared to traditional plankton sampling discussed above. In contrast to the HSN net, which has a smaller mouth opening leading to a smaller sampling volume, the Manta net benefits from a larger opening and lower towing speed. This combination reduces turbulence and allows for a larger sampling volume, resulting in potentially lower loss. This is reflected in Fig. 7a, where the Manta net captures a wider range of sizes, including larger and rarer fragile organisms. Skjoldal et al. (2019) measured less biomass in the large size fraction and more biomass in the small and medium size fractions at the higher towing speeds. The opposite effect might have been expected for the small fraction due to extrusion (Skjoldal et al., 2019), suggesting that the HSN net may be more effective at capturing smaller organisms. However, this is not clearly demonstrated, as the slopes of the HSN's NBSS are largely equivalent to those of the Manta (mean NBSS slope for HSN = -0.35, std = 0.30 and mean NBSS slope for Manta = -0.30, std = 0.23; Fig. 7a). This also suggests that both nets capture the same trophic plankton ecosystem structure, while the HSN underestimates plankton in each size class.

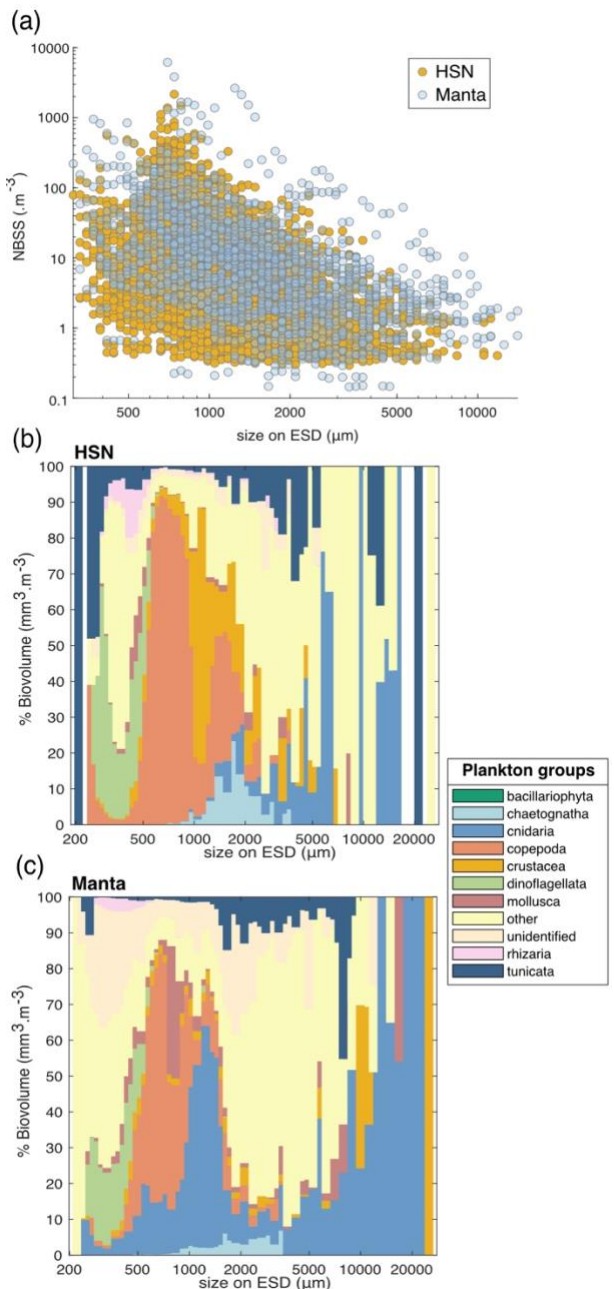

**Figure 7. (a) Comparison of Normalized Biovolume Size Spectra (NBSS) of living organisms sampled with HSN in yellow dots and Manta nets in blue dots. Only stations where both were deployed are included in this figure. Average taxonomic composition of the 'plankton groups' in biovolume (mm³/m⁻³) for all stations by size class (in µm) for samples collected with HSN in (b) and Manta net in (c).**

All these observed differences may therefore introduce differences in species composition. Investigating the taxonomic composition, the HSN and the Manta show on average relatively similar community compositions (Fig. 7c and 7d; the dinoflagellates are almost entirely composed of the genus Noctiluca). Investigating the taxonomic composition in terms of biovolume, the five most represented groups in the Manta dataset are Cnidaria (59%), Copepoda (13%), other (11%), Crustacea (9%), and Mollusca (3%). In contrast, the HSN dataset shows a more even distribution, with other taxa contributing 33%, followed by Cnidaria (28%), Copepoda (19%), Tunicata (10%), and Crustacea (6%). Although there is a general difference in the sampled plankton community, the greatest discrepancies are observed for gelatinous organisms. Thus, HSN net undersampled larger and more fragile

organisms such as cnidarians and tunicates (Fig. 7c). This aligns with the limitations of high-speed deployments, which have been shown to damage delicate organisms (Harris et al., 2000; Keen, 2013). This damage to large and fragile plankton could cause the higher concentrations of smaller size classes we found in HSN compared to Manta samples. In contrast, the HSN consistently sampled more robust organisms such as copepods and chaetognaths than the Manta (Fig. 7c and 6d).

For the quantitative and qualitative comparison of plankton community sampling, we only considered stations where both nets were deployed sequentially (first the Manta, followed by the HSN). Although small, this temporal and spatial difference remains a limitation in our comparison between the two nets. In terms of location, this combination of Manta-HSN deployments was primarily conducted near islands, where plankton concentrations and composition are known to be highly variable (Bourdin et al., 2024; Kristan et al., in prep). Given that the Manta was deployed before the HSN, i.e., closer to the islands, we also expect part of the HSN underestimation signal to be explained by this small spatial difference. Therefore, while our primary hypothesis attributes these differences mainly to the high-speed deployment of the HSN (up to three times greater than that of the Manta), these spatial and temporal factors, in addition to the patchiness distribution of plankton (Robinson et al., 2021), may also play a role in our comparison of the two plankton sampling systems.

4. General discussion

In conclusion to our investigation of sampling biases associated with the high-speed sampling, the HSN must therefore be considered as semi-quantitative. The use of the HSN introduces an undersampling bias that is also found in other high-speed samplers, as described for the CPR. Nevertheless, we highlight the usefulness of the HSN for sampling surface zooplankton when it is not possible to stop or slow the boat, and its value in extending sampling coverage and frequency. Consistent with the CPR, HSN captures a roughly consistent fraction of the in-situ abundance reflecting the main patterns observed in plankton. Consistent with expected ecological trends, higher plankton abundances and biovolumes are observed in nutrient-rich regions such as coastal and upwellings, whereas oligotrophic gyres exhibit significantly lower biomass (see abundance, biovolume, and diversity maps for each sampling device in appendix B). For example, the trend of increasing plankton abundance due to California upwelling (Checkley and Barth, 2009) appears to emerge regardless of the sampling method used (appendix B: Fig. B1 to B4). Each net is a filter through which we sample the ocean, but if the overall patterns they show are consistent, we can conclude that they are likely to be robust patterns. This is true for many types of sampling nets, as many previous studies have shown (Herdman, 1921; Barnes and Marshall, 1951; Anraku, 1956; Wiebe and Holland, 1968).

In addition to the unique characteristic of high-speed sampling, these datasets are also distinguished by their focus on surface plankton communities during daytime, offering both advantages and limitations. These surface plankton data enrich interdisciplinary studies of ocean's surface layer, in direct associations with other surface measurements (satellite and atmospheric data; Lombard et al., 2019). This surface ecosystem, hosting a uniquely diverse planktonic community, remains largely unexplored, but appears to play an essential role in ocean-climate feedbacks (Helm, 2021; Hunter, 2023) as a critical interface between atmospheric and oceanic process and contributing significantly to biogeochemical cycles (Falkowski et al., 2008). Processes controlling the abundance and diversity of the surface plankton communities may be significantly different from those in deeper layers (Ibarbalz et al., 2019, Santiago et al.,2023). The surface is also on the frontline of climate change and pollution. Thus, these particular communities face increasing challenges such as rising temperatures, stratification and nutrient stress (Bopp et al., 2013; IPCC, 2022) and floating contaminants ranging from plastics, metals and toxins to petroleum (Helm, 2021). However, surface plankton sampling has limitations regarding the "quantitative representativeness" of the broader plankton ecosystem in the water column. The Tara Pacific sampling was conducted under stable daytime conditions, minimizing variability from diel vertical migration (Lampert, 1989). As a result, zooplankton concentrations do not reflect deeper-dwelling organisms, particularly those migrating to the surface at night, leading to potentially higher abundances within the water column (Lampert, 1989). This is also valuable for phytoplankton communities that are known to be heterogeneously distributed from the surface

to deeper waters into the euphotic zone, especially in the transparent oligotrophic waters of the Pacific gyre, where Deep Chlorophyll Maxima can occur tens to hundreds of meters below the surface (Mignot et al., 2014). In terms of comparison with non-surface plankton data, this limitation must be carefully considered by future users.

**Conclusion**

The Tara Pacific Expedition is part of the first initiatives aiming to implement a system for discrete sampling of the planktonic ecosystem while operating at cruising speed (5–9 knots), covering viruses to metazoa at the scale of the whole expedition (Gorsky et al., 2019) and focusing on micro- to mesoplankton in this paper. The use of two new sampling systems highlights some biases that lead to undersampling, which is important to consider in subsequent ecological analyses. However, the simultaneous high-speed sampling of the different components of the surface ecosystem may contribute to address the issue of undersampling of the open ocean at difficult-to-reach spatial and temporal scales, a major challenge for marine science. These systems can be improved and adapted to vessels of different sizes and propulsion systems, opening the way to complementary initiatives, such as plankton collection by citizen sailors. (De Vargas et al., 2022; Mériguet et al., 2022).

In conclusion, using these new sampling methods covering the North and South Pacific and North Atlantic basins, we provide an important dataset focusing on the surface plankton rarely sampled as a whole. Our large-scale analysis reveals an important taxonomic and functional diversity within the surface planktonic communities, encompassing approximately 370 different taxa, primarily identified at the genus level, spanning across 12 major plankton groups and 5 trophic levels. We hope that the dataset presented here, will stimulate further studies (i.e., biodiversity, biogeochemistry, modeling studies…) using the different environmental imprints recorded during the Tara Pacific expedition (data available in Lombard et al., 2023) to highlight the processes influencing this particular plankton ecosystem, from large scale to mesoscale levels, from taxonomic scale to trophic scale, or from species barcodes to genomes. Such an important dataset will not only serve as a starting point for many studies to deepen our understanding of planktonic ecosystems, their biogeochemical roles, and their socio-economic importance, but could also serve as a reference state of the ecosystem in the context of environmental changes.

**4. Data availability**

The referenced datasets related to figures are available at:
https://doi.org/10.17882/102537 Mériguet et al., (2024a) (EcoTaxa link: https://ecotaxa.obs-vlfr.fr/prj/1344 and https://ecotaxa.obs-vlfr.fr/prj/1345),
https://doi.org/10.17882/102336 Mériguet et al., (2024b) (EcoTaxa link: https://ecotaxa.obs-vlfr.fr/prj/11292),
https://doi.org/10.17882/102694 Mériguet et al., (2024c) (EcoTaxa link: https://ecotaxa.obs-vlfr.fr/prj/11370 and https://ecotaxa.obs-vlfr.fr/prj/11369)
and https ://doi.org/10.17882/102697 Mériguet et al., (2024d) (EcoTaxa link: https://ecotaxa.obs-vlfr.fr/prj/11353 and https://ecotaxa.obs-vlfr.fr/prj/11341).

The imaging datasets are also summarized in Table 2.

A key strength of this quantitative imaging dataset is its complementarity with a wide range of environmental data collected during the Tara Pacific expedition. This expedition is described in detail in Lombard et al. (2023), where the full set of environmental datasets is available and referenced: https://doi.org/10.1038/s41597-022-01757-w. Environmental data were collected station by station, making it possible to link them directly to our dataset using the station name. Each station is identified by a unique [oa###] code, where the "oa" label is the key identifier for associating environmental measurements with our imaging data. When looking at data at this 'station' level, all environmental data are already compiled and compatible for easy analysis and cross-analysis, and when linked to sample barcodes, they could be further linked to any other associated data (e.g. genomic) by linking them to the sample registry available in Lombard et al 2023, with sample and event registry at:

https://doi.org/10.1594/PANGAEA.944548. In addition to station-based data, continuous environmental measurements from the Tara Pacific expedition (Lombard et al., 2023) can also be linked to our dataset. These measurements can be linked to plankton net sampling events using date, time and GPS coordinates, all of which are available in both the plankton and in line environmental datasets. This ensures a robust integration of imaging and environmental data, facilitating large-scale ecological analyses.

| | **Datasets** | | | |
|---|---|---|---|---|
| **Name** | **FlowCam Tara Pacific DN 20 microns** | **FlowCam Tara Pacific Bongo 20 microns** | **ZooScan Tara Pacific HSN 330 microns** | **ZooScan Tara Pacific Manta 333 microns** |
| **DOI** | 10.17882/102697 | 10.17882/102694 | 10.17882/102336 | 10.17882/102537 |
| **Sampling Location** | Open-ocean and islands sampling | Islands, reef and lagoon sampling | Open-ocean and islands sampling | Open-ocean (Great Pacific Garbage Patch) and islands sampling |
| **Plankton size imaged** | (20-200 μm) | (20-200 μm) | (> 300 μm) | (> 300 μm) |
| **Link to open EcoTaxa project** | Subset 30% < 500 pixels:<br><br>https://ecotaxa.obs-vlfr.fr/prj/11353<br><br>Subset 100 % > 501 pixels:<br><br>https://ecotaxa.obs-vlfr.fr/prj/11341 | Subset 30% < 500 pixels:<br><br>https://ecotaxa.obs-vlfr.fr/prj/11370<br><br>Subset 100 % > 501 pixels:<br><br>https://ecotaxa.obs-vlfr.fr/prj/11369 | https://ecotaxa.obs-vlfr.fr/prj/11292 | Subset Plankton images<br><br>https://ecotaxa.obs-vlfr.fr/prj/1344<br><br>Subset Plastics images<br><br>https://ecotaxa.obs-vlfr.fr/prj/1345 |
| **ZIP files with one tsv per samples, raw export from EcoTaxa** | Subset 30% < 500 pixels:<br><br>Export EcoTaxa FlowCam Tara Pacific DN 20 microns < 500 pixels.zip<br><br>Subset 100 % > 501 pixels:<br><br>Export EcoTaxa FlowCam Tara Pacific DN 20 microns > 501 pixels.zip | Subset 30% < 500 pixels:<br><br>Export EcoTaxa FlowCam Tara Pacific Bongo 20 microns < 500 pixels.zip<br><br>Subset 100 % > 501 pixels:<br><br>Export EcoTaxa FlowCam Tara Pacific Bongo 20 microns > 501 pixels.zip | Export EcoTaxa ZooScan Tara Pacific HSN 330 microns.zip | Subset Plankton images<br><br>Export EcoTaxa ZooScan Tara Pacific Manta 333 microns plankton.zip<br><br>Subset Plastics images<br><br>Export EcoTaxa ZooScan Tara Pacific Manta 333 microns plastics.zip |

| CSV files with ab, bv (x3: area, riddled and ellispoidal), shannon | Descriptors FlowCam Tara Pacific DN 20 microns.csv | Descriptors FlowCam Tara Pacific Bongo 20 microns.csv | Descriptors ZooScan Tara Pacific HSN 330 microns.csv | Descriptors ZooScan Tara Pacific Manta 333 microns.csv |
|---|---|---|---|---|
| ZIP files with 1 table csv / sample for NBSS (1 NBSS / sample) | NBSS FlowCam Tara Pacific DN 20 microns.zip | NBSS FlowCam Tara Pacific Bongo 20 microns.zip | NBSS ZooScan Tara Pacific HSN 330 microns.zip | NBSS ZooScan Tara Pacific Manta 333 microns.zip |

**Table 2. Summary of data availability, description and useful link for each dataset.**

**Appendices**

| FlowCam Tara Pacific DN 20 microns | | |
|---|---|---|
| Taxonomic list | Plankton groups | Trophic type |
| Bacillariophyceae | bacillariophyta | phototroph |
| Asterionellopsis | | |
| Asterolamprales | | |
| Bacillariaceae | | |
| Climacodium | | |
| Climacodium inter. Crocosphaera | | |
| chainlarge | | |
| chainthin | | |
| multiple < Diatoma | | |
| Pseudo-Nitzschia chain | | |
| Thalassionematales | | |
| Corethron | | |
| Coscinodiscophycidae | | |
| Coscinodiscids | | |
| Bacteriastrum | | |
| Chaetoceros | | |
| Chaetoceros protuberans | | |
| Chaetoceros peruvianus | | |
| Ditylum | | |
| Eucampia | | |
| Hemiaulus | | |
| Fragilariopsis | | |
| Nitzschia | | |
| Planktoniella sol | | |
| Rhizosolenids | | |

| | | |
|---|---|---|
| Dactyliosolen | | |
| Guinardia | | |
| Rhizosolenia inter. Richelia | | |
| pennate < Bacillariophyta | | |
| Helicotheca | | |
| | | |
| Cyanobacteria | cyanobacteria | autotroph |
| UCYNA like | | |
| cyano a | | |
| cyano b | | |
| Richelia | | |
| attached | | |
| | | |
| Codonaria | ciliophora | mixotroph |
| Ciliophora | | |
| Amphorides | | |
| Codonellidae | | |
| Codonellopsis | | |
| Codonellopsis orthoceras | | |
| Cyttarocylis | | |
| Dictyocysta | | |
| Epiplocylis | | |
| Eutintinnus | | |
| Lacrymaria | | |
| Metacylis | | |
| Poroecus | | |
| Rhabdonella | | |
| Rhabdonellopsis | | |
| Salpingella | | |
| Steenstrupiella | | |
| Tintinnida | | |
| Undellidae | | |
| Amplectella | | |
| Xystonellidae | | |
| Dadayiella | | |
| Zoothamniidae | | |
| | | |
| Dictyochophyceae | dictyochophyceae | phototroph |
| | | |
| Gonyaulacales | dinoflagellata | mixotroph |
| Dinophyceae | | |
| Amphisolenia | | |
| Dinophysis | | |

| | | |
|---|---|---|
| Ceratocorys | | |
| Cladopyxis | | |
| Neoceratium | | |
| Neoceratium limulus | | |
| Neoceratium candelabrum | | |
| Neoceratium furca | | |
| Neoceratium fusus | | |
| Neoceratium pentagonum | | |
| Neoceratium geniculatum | | |
| Pyrocystaceae | | |
| Pyrophacus | | |
| Gymnodiniales | | |
| Ornithocercus | | |
| Ornithocercus heteroporus | | |
| Ornithocercus magnificus | | |
| Ornithocercus quadratus | | |
| Ornithocercus steinii | | |
| Oxytoxum | | |
| Phalacroma | | |
| Podolampas | | |
| Protoperidinium | | |
| polar view | | |
| Hemidiscus cuneiformis | | |
| | | |
| Tunicata | tunicata | grazers |
| Appendicularia | | |
| Copepoda | copepoda | |
| Ostracoda | crustacea | |
| nauplii < Crustacea | | |
| Rotifera | other | |
| trochozoa | | |
| larvae < Annelida | | omnivorous |
| | | |
| veliger | mollusca | grazers |
| | | |
| Pterosperma | other | phototroph |
| | | |
| Rhizaria | rhizaria | mixotroph |
| Retaria | | |
| Amphibelone | | |
| Acantharia | | |
| Foraminifera | | |
| Nassellaria | | |

| Taxonomic list | Plankton groups | Trophic type |
|---|---|---|
| Spumellaria | | |
| | | |
| cyst | | |
| egg | other | – |
| egg sac | | |
| | | |
| multiple < other | – | – |
| | | |
| othertocheck | | |
| darkrods < othertocheck | other unidentified | unidentified |
| lightrods < othertocheck | | |
| othersphere | | |
| | | |
| t001 | | |
| t003 | other unidentified | unidentified |
| t004 | | |
| | | |
| tail < Appendicularia | | |
| part < Crustacea | | |
| spines < Acantharea | | |
| part < Ciliophora | | |
| artefact | | |
| badfocus < artefact | | |
| bubble | | |
| detritus | non-living | – |
| dark < detritus | | |
| fiber < detritus | | |
| light < detritus | | |
| pollen | | |
| duplicate | | |
| t002 | | |

**Table A1. List of EcoTaxa taxonomic annotations and associated groups: plankton groups and trophic type for the**
**FlowCam DN 20 microns dataset.**

| FlowCam Tara Pacific Bongo 20 microns | | |
|---|---|---|
| Taxonomic list | Plankton groups | Trophic type |
| Trichodesmium | | |
| UCYNA like | cyanobacteria | autotroph |
| Cyanobacteria<Proteobacteria | | |
| Richelia | | |
| | | |
| Ciliophora | ciliophora | mixotroph |

| | | |
|---|---|---|
| Lacrymaria<Lacrymariidae | | |
| Vorticella | | |
| Codonellidae | | |
| Cyttarocylis | | |
| Epiplocylis | | |
| Dictyocysta | | |
| Metacylis | | |
| Rhabdonella | | |
| Rhabdonellopsis | | |
| Tintinnida | | |
| tintinnid-diatom | | |
| Amphorides<Tintinnidiidae | | |
| Eutintinnus | | |
| Salpingella<Tintinnidiidae | | |
| Steenstrupiella | | |
| Tintinnidae X | | |
| Poroecus | | |
| Undellidae | | |
| Xystonellidae | | |
| part<Ciliophora | | |
| | | |
| Dinophyceae | | |
| Dinophyceae X | | |
| Amphisolenia | | |
| Ornithocercus | | |
| Ornithocercus magnificus<Ornithocercus | | |
| Ornithocercus steinii | | |
| Phalacroma | | |
| Neoceratium | | |
| Neoceratium candelabrum | | |
| Neoceratium furca<Neoceratium | dinoflagellata | mixotroph |
| Neoceratium fusus<Neoceratium | | |
| Neoceratium pentagonum | | |
| Cladopyxis | | |
| Ostreopsis | | |
| Pyrocystaceae | | |
| Pyrophacus | | |
| Peridiniales | | |
| Oxytoxum | | |
| Podolampas | | |
| Protoperidinium | | |

| | | |
|---|---|---|
| Rhizaria | | |
| Retaria | | |
| Acantharea | | |
| spines<Acantharea | | |
| Foraminifera | | |
| Nassellaria<Polycystinea | rhizaria | mixotroph |
| Spumellaria | | |
| Radiolaria | | |
| aggregate<Radiolaria | | |
| part<Rhizaria | | |
| spines<Rhizaria | | |
| | | |
| Bacillariophyceae | | |
| Asterionella | | |
| Coscinodiscophycidae | | |
| Asterolamprales | | |
| Hemidiscus cuneiformis | | |
| Hemidiscus | | |
| Cylindrotheca | | |
| Diatoma | | |
| chainlarge | | |
| chainthin | | |
| multiple<Diatoma | | |
| Licmophora | | |
| Naviculales | | |
| Nitzschia | | |
| Pseudo-nitzschia | bacillariophyta | phototroph |
| Striatella | | |
| Synedra | | |
| Thalassionematales | | |
| Amphitetras | | |
| Bacteriastrum<Mediophyceae | | |
| Biddulphia | | |
| Chaetoceros<Mediophyceae | | |
| Chaetoceros inter ciliate | | |
| Chaetoceros inter. Calothrix | | |
| Ditylum | | |
| Eucampia | | |
| Hemiaulus | | |
| Odontella sp. | | |

| | | |
|---|---|---|
| Odontella<Mediophyceae | | |
| Planktoniella | | |
| Corethron | | |
| Coscinodiscus | | |
| Stephanopyxis | | |
| Rhizosolenids | | |
| Dactyliosolen | | |
| Guinardia | | |
| Rhizosolenia | | |
| Rhizosolenia inter. Richelia | | |
| rhizosolenia inter richelia tmp i | | |
| rhizosolenia tmp i | | |
| centric | | |
| chain<centric | | |
| pennate<Bacillariophyta | | |
| part diatom | | |
| | | |
| Dictyochophyceae | dictyochophyceae | phototroph |
| Dictyochales | | |
| Dictyocha | | |
| | | |
| Annelida | others | grazers |
| larvae<Polychaeta | | |
| trocophora | | |
| larvae<Annelida | | |
| trochophore | | |
| | | |
| Copepoda<Maxillopoda | copepoda | omnivorous |
| Calanoida | | |
| Cyclopoida | | |
| Oithonidae | | |
| Harpacticoida | | |
| Corycaeidae | | |
| Oncaeidae | | |
| part<Copepoda | | |
| | | |
| nauplii<Crustacea | crustacea | grazers |
| part<Crustacea | | |
| | | |
| Bryozoa | other | grazers |
| trochozoa | | |

| | | |
|---|---|---|
| larvae<Echinodermata | | |
| Mollusca | mollusca | |
| veliger | | |
| | | |
| larvae<living | other | unidentified |
| other<living | | |
| egg<other | | – |
| egg sac<egg | | |
| | | |
| multiple<other | – | – |
| duplicate | | |
| | | |
| othertocheck | other unidentified | unidentified |
| crumple sphere | | |
| darkrods<othertocheck | | |
| lightrods<othertocheck | | |
| | | |
| t001 | other unidentified | unidentified |
| t002 | | |
| t003 | | |
| t004 | | |
| t005 | | |
| t006 | | |
| t007 | | |
| t008 | | |
| t010 | | |
| t011 | | |
| t012 | | |
| t013 | | |
| t014 | | |
| t015 | | |
| t016 | | |
| t017 | | |
| | | |
| part<other | non-living | – |
| part<seaweed | | |
| Micracanthodinium quadrispinum | | |
| artefact | | |
| badfocus<artefact | | |
| bubble | | |
| detritus | | |

| aggregates |
|---|
| dark<detritus |
| fiber<detritus |
| light<detritus |
| feces |
| darkrods<rods |
| lightrods<rods |

**Table A2. List of EcoTaxa taxonomic annotations and associated groups: plankton groups and trophic type for the**
**FlowCam Bongo 20 microns dataset.**

| ZooScan Tara Pacific HSN 330 microns | | |
|---|---|---|
| Taxonomic list | Plankton groups | Trophic type |
| Actinopterygii | other | predators |
| egg < Actinopterygii | | |
| | | |
| Annelida | other | omnivorous |
| Spirorbis | | |
| larvae < Annelida | | |
| | | |
| Appendicularia | tunicata | grazers |
| Oikopleuridae | | |
| | | |
| Bryozoa | other | grazers |
| cyphonaute | | |
| | | |
| Chaetognatha | chaetognatha | predators |
| | | |
| Hydrozoa | cnidaria | predators |
| Scyphozoa | | |
| Porpita | | |
| larvae < Porpitidae | | |
| Siphonophorae | | |
| bract < Abylidae | | |
| gonophore < Abylidae | | |
| nectophore < Abylidae | | |
| Diphyidae | | |
| bract < Diphyidae | | |
| eudoxie < Diphyidae | | |
| gonophore < Diphyidae | | |
| nectophore < Diphyidae | | |
| nectophore < Hippopodiidae | | |

| | | |
|---|---|---|
| Abylopsis tetragona | | |
| bract < Abylopsis tetragona | | |
| eudoxie < Abylopsis tetragona | | |
| gonophore < Abylopsis tetragona | | |
| nectophore < Abylopsis tetragona | | |
| bract < Bassia bassensis | | |
| nectophore < Bassia bassensis | | |
| Physonectae | | |
| nectophore < Physonectae | | |
| Velella | | |
| polype < Leptothecata | | |
| polype < Anthozoa | | |
| | | |
| Cirripedia | | |
| cirrus | | |
| cypris | crustacea | grazers |
| nauplii < Cirripedia | | |
| Evadne | | |
| Podon | | |
| | | |
| Calanoida | | |
| Acartiidae | | |
| Calanidae | | |
| Calocalanus pavo | | |
| Candaciidae | | |
| Centropagidae | | |
| Eucalanidae | | |
| Euchaetidae | | |
| Heterorhabdidae | | |
| Metridinidae | | |
| Pontellidae | copepoda | omnivorous |
| Pontellina plumata | | |
| Monstrilloida | | |
| Temoridae | | |
| Oithonidae | | |
| Harpacticoida | | |
| Corycaeidae | | |
| Oncaeidae | | |
| Sapphirinidae | | |
| Copilia | | |
| Lubbockia | | |

| | | |
|---|---|---|
| Siphonostomatoida | | |
| badfocus < Copepoda | | |
| damaged < Copepoda | | |
| multiple < Copepoda | | |

| | | |
|---|---|---|
| Crustacea | crustracea | predators |
| Eumalacostraca | | |
| Amphipoda | | |
| Caprellidae | | |
| Gammaridea | | |
| protozoea | | |
| Hyperiidea | | |
| Brachyura | | |
| Phronimidae | | |
| megalopa | | |
| zoea < megalopa | | |
| Euphausiacea | | |
| calyptopsis < Euphausiacea | | |
| Isopoda | | |
| Laomediidae | | |
| larvae < Porcellanidae | | |
| phyllosoma | | |

| | | |
|---|---|---|
| nauplii < Crustacea | crustracea | grazers |
| metanauplii < Crustacea | | |
| Ostracoda | | |
| larvae < Squillidae | | |

| | | |
|---|---|---|
| Cyanobacteria < Bacteria | cyanobacteria | autotroph |

| | | |
|---|---|---|
| Echinodermata | other | grazers |
| echinopluteus | | |
| pluteus < echinoidea | | |
| ophiuroidea | | |
| ophiopluteus | | |
| pluteus<ophioroidea | | |

| | | |
|---|---|---|
| Harosa | rhizaria | mixotroph |
| Acantharia | | |
| Collodaria | | |
| Globorotalidae | | |

| | | |
|---|---|---|
| Orbunila | | |
| Foraminifera | | |
| Spumellaria | | |
| | | |
| Pyrocystaceae | dinoflagellata | mixotroph |
| multiple < Pyrocystaceae | | |
| | | |
| Insecta | other | predators |
| Halobates | | |
| | | |
| Mollusca | mollusca | grazers |
| Bivalva | | |
| Gymnosomata | | |
| Cavolinia inflexa | | |
| Diacria | | |
| Atlanta | | |
| Cavoliniidae | | |
| Cephalopoda | | |
| Creseidae | | |
| Creseis acicula | | |
| Creseis virgula | | |
| Firola | | |
| Limacinidae | | |
| part < Mollusca | | |
| veliger | | |
| | | |
| Doliolida | tunicata | predators |
| Salpida | | |
| juvenil < Salpida | | |
| nucleus < Salpida | | |
| | | |
| egg < other | other | _ |
| egg sac < egg | | |
| | | |
| gelatinous | other | predators |
| | | |
| nudibranchia | other | _ |
| | | |
| multiple < other | other | _ |
| | | |
| othertocheck | other unidentified | unidentified |

| | | |
|---|---|---|
| darksphere | | |
| othersphere | | |
| | | |
| t001 | | |
| t002 | other unidentified | unidentified |
| t003 | | |
| t004 | | |
| | | |
| part < Actinopterygii | | |
| scale < Actinopterygii | | |
| trunk < Appendicularia | | |
| head < Chaetognatha | | |
| part < Annelida | | |
| tail < Appendicularia | | |
| tail < Chaetognatha | | |
| part < Thaliacea | | |
| part < Siphonophorae | | |
| part < Copepoda | | |
| part < Cnidaria | non-living | _ |
| part < Crustacea | | |
| part < Ctenophora | | |
| wing < Halobates | | |
| empty < Ostracoda | | |
| artefact | | |
| badfocus < artefact | | |
| bubble | | |
| detritus | | |
| borax | | |
| dark < detritus | | |
| fiber < detritus | | |

**Table A3. List of EcoTaxa taxonomic annotations and associated groups: plankton groups and trophic type for the**
**ZooScan HSN 330 microns dataset.**

| Tara Pacific 2016 2018 Manta 300 plankton | | |
|---|---|---|
| Taxonomic list | Plankton groups | Trophic type |
| Actinopterygii | other | predators |
| egg < Actinopterygii | | |
| | | |
| Annelida | other | omnivorous |
| larvae < Annelida | | |

| | | |
|---|---|---|
| Alciopidae | | |
| Tomopteridae | | |
| Spirorbis | | |
| Terebellidae | | |
| | | |
| Fritillariidae | tunicata | grazers |
| Oikopleuridae | | |
| | | |
| Chaetognatha | chaetognatha | predators |
| | | |
| Cnidaria | cnidaria | predators |
| polype < Anthozoa | | |
| Hydrozoa | | |
| larvae < Porpitidae | | |
| Porpita porpita | | |
| Velella | | |
| polype < Leptothecata | | |
| bract < Abylopsis tetragona | | |
| eudoxie < Abylopsis tetragona | | |
| gonophore < Abylopsis tetragona | | |
| nectophore < Abylopsis tetragona | | |
| bract < Bassia bassensis | | |
| gonophore < Bassia bassensis | | |
| nectophore < Bassia bassensis | | |
| bract < Diphyidae | | |
| Chelophyes | | |
| eudoxie < Diphyidae | | |
| eudoxie < Eudoxoides spiralis | | |
| gonophore < Eudoxoides spiralis | | |
| nectophore < Eudoxoides spiralis | | |
| gonophore < Diphyidae | | |
| nectophore < Diphyidae | | |
| nectophore < Hippopodiidae | | |
| Physalia | | |
| nectophore < Physonectae | | |
| Aglaura | | |
| Rhopalonema velatum | | |
| ephyra | | |
| | | |
| Ctenophora | other | predators |
| | | |

| | | |
|---|---|---|
| cirrus | crustacea | grazers |
| cypris | | |
| nauplii < Cirripedia | | |
| Evadne | | |
| larvae < Crustacea | | |
| metanauplii < Crustacea | | |
| | | |
| Eumalacostraca | crustacea | predators |
| Amphipoda | | |
| Gammaridea | | |
| Hyperiidea | | |
| Oxycephalidae | | |
| Phronima | | |
| protozoea < Penaeidae | | |
| protozoea < Sergestidae | | |
| zoea < Galatheidae | | |
| larvae < Porcellanidae | | |
| Brachyura | | |
| megalopa | | |
| zoea < Brachyura | | |
| like < Laomediidae | | |
| calyptopsis | | |
| protozoea < Mysida | | |
| | | |
| Crustacea | crustacea | predators |
| nauplii < Crustacea | | |
| metanauplii < Crustacea | | |
| Ostracoda | | |
| larvae < Squillidae | | grazers |
| | | |
| Copepoda | copepoda | omnivorous |
| Calanoida | | |
| Acartiidae | | |
| Haloptilus | | |
| Calanidae | | |
| Candaciidae | | |
| Centropagidae | | |
| Eucalanidae | | |
| Euchaetidae | | |
| Metridinidae | | |
| Calocalanus pavo | | |

| | | |
|---|---|---|
| Pontellidae | | |
| Pontellina plumata | | |
| Temoridae | | |
| Oithonidae | | |
| Harpacticoida | | |
| Miraciidae | | |
| Corycaeidae | | |
| Lubbockia | | |
| Oncaeidae | | |
| Sapphirinidae | | |
| Copilia | | |
| badfocus < Copepoda | | |
| multiple < Copepoda | | |
| damaged < Copepoda | | |
| | | |
| Insecta | other | predators |
| Gerridae | | |
| | | |
| Bryozoa | other | grazers |
| cyphonaute | | |
| | | |
| Branchiostoma lanceolatum | other | grazers |
| | | |
| Doliolida | tunicata | omnivorous |
| Pyrosomatida | | |
| Salpida | | |
| chain < Salpida | | |
| juvenile < Salpida | | |
| | | |
| Mollusca | mollusca | grazers |
| Bivalvia | | |
| Cephalopoda | | |
| Atlanta | | |
| Firola | | |
| Gymnosomata | | |
| Cavoliniidae | | |
| Diacavolinia | | |
| Diacria trispinosa | | |
| Creseidae | | |
| Creseis acicula | | |
| Creseis virgula | | |

| | | |
|---|---|---|
| Limacinidae | | |
| Nudibranchia | | |
| egg < Mollusca | other | _ |
| | | |
| pluteus < Echinoidea | other | omnivorous |
| pluteus < Ophiuroidea | | |
| | | |
| Harosa | other | |
| Neoceratium | dinoflagellata | |
| Pyrocystaceae | | mixotroph |
| Foraminifera | | |
| Orbulina | rhizaria | |
| Spumellaria | | |
| Diatoma | diatoms | phototroph |
| | | |
| egg < other | other | _ |
| | | |
| living < other | other | _ |
| | | |
| multiple < other | other | _ |
| | | |
| othertocheck | other unidentified | unidentified |
| | | |
| seaweed | other | phototroph |
| | | |
| t002 | other unidentified | unidentified |
| t003 | | |
| t004 | | |
| t005 | | |
| t007 | | |
| t008 | | |
| t010 | | |
| t012 | | |
| t013 | | |
| t014 | | |
| t015 | | |
| t016 | | |
| t017 | | |
| | | |
| plastic<fiber | plastics | _ |
| plastic<filament | | |

| | | |
|---|---|---|
| plastic<film | | |
| plastic<fragment | | |
| plastic<multiple | | |
| plastic<other | | |
| plastic<pellet | | |
| plastic<polystyrene | | |
| | | |
| part<Copepoda | | |
| part<other | | |
| scale<Actinopterygii | | |
| part<Annelida | | |
| tail<Appendicularia | | |
| trunk<Appendicularia | | |
| head<Chaetognatha | | |
| tail<Chaetognatha | | |
| part<Siphonophorae | | |
| part<Cnidaria | | |
| part<Ctenophora | | |
| part<Crustacea | non-living | – |
| wing<Insecta | | |
| part<Thaliacea | | |
| nucleus<Salpida | | |
| part<Mollusca | | |
| detritus | | |
| artefact | | |
| badfocus<artefact | | |
| bubble | | |
| dark<detritus | | |
| fiber<detritus | | |

**Table A4. List of EcoTaxa taxonomic annotations and associated groups: plankton groups and trophic type for the**
**ZooScan Manta 333 microns dataset.**

**DN-FlowCam**

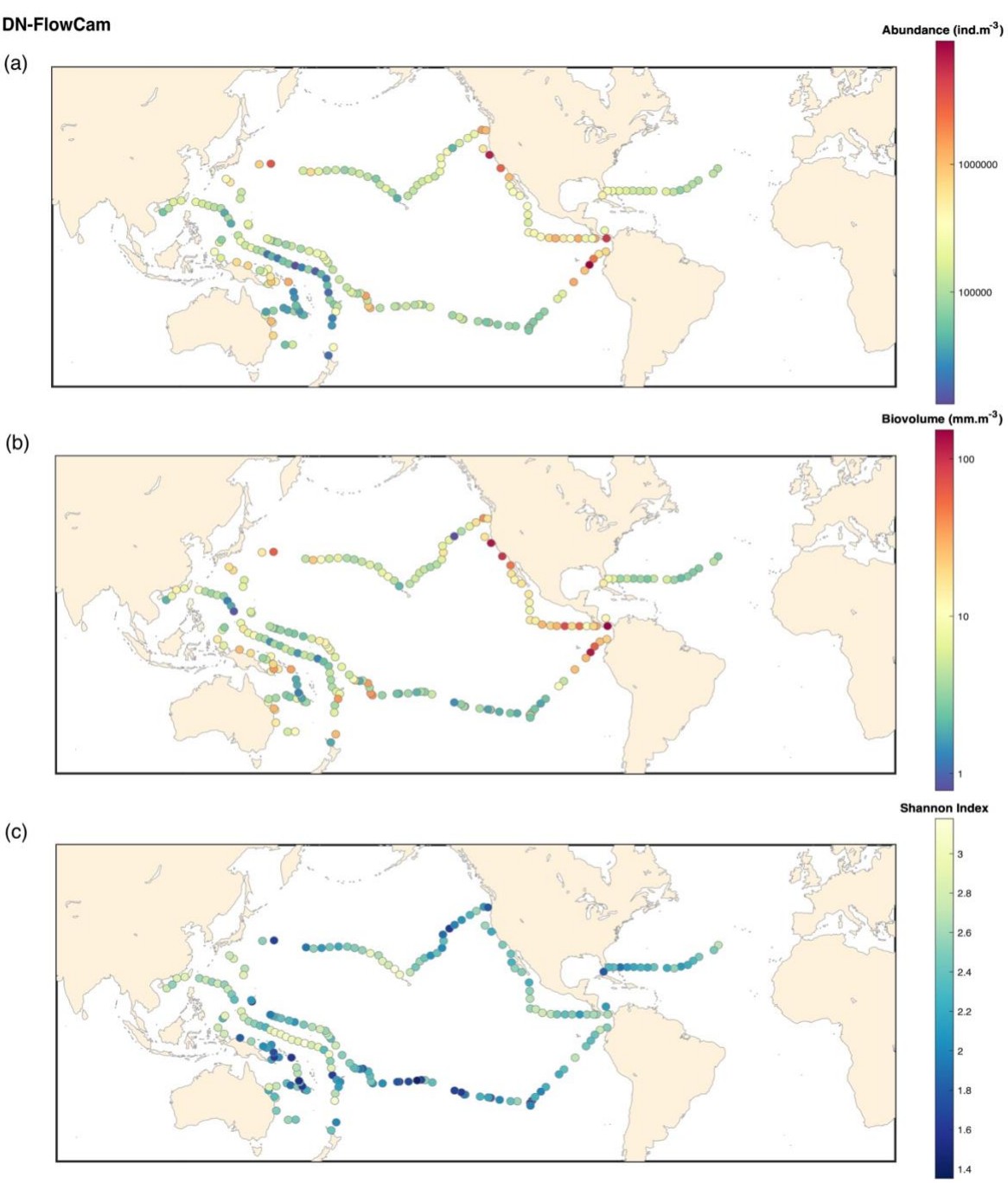

Figure B1. FlowCam DN 20 microns: (a) Map of plankton abundance (ind.m$^{-3}$). (b) Map of plankton biovolume (mm.m$^{-3}$). (c) Map of Shannon diversity Index.

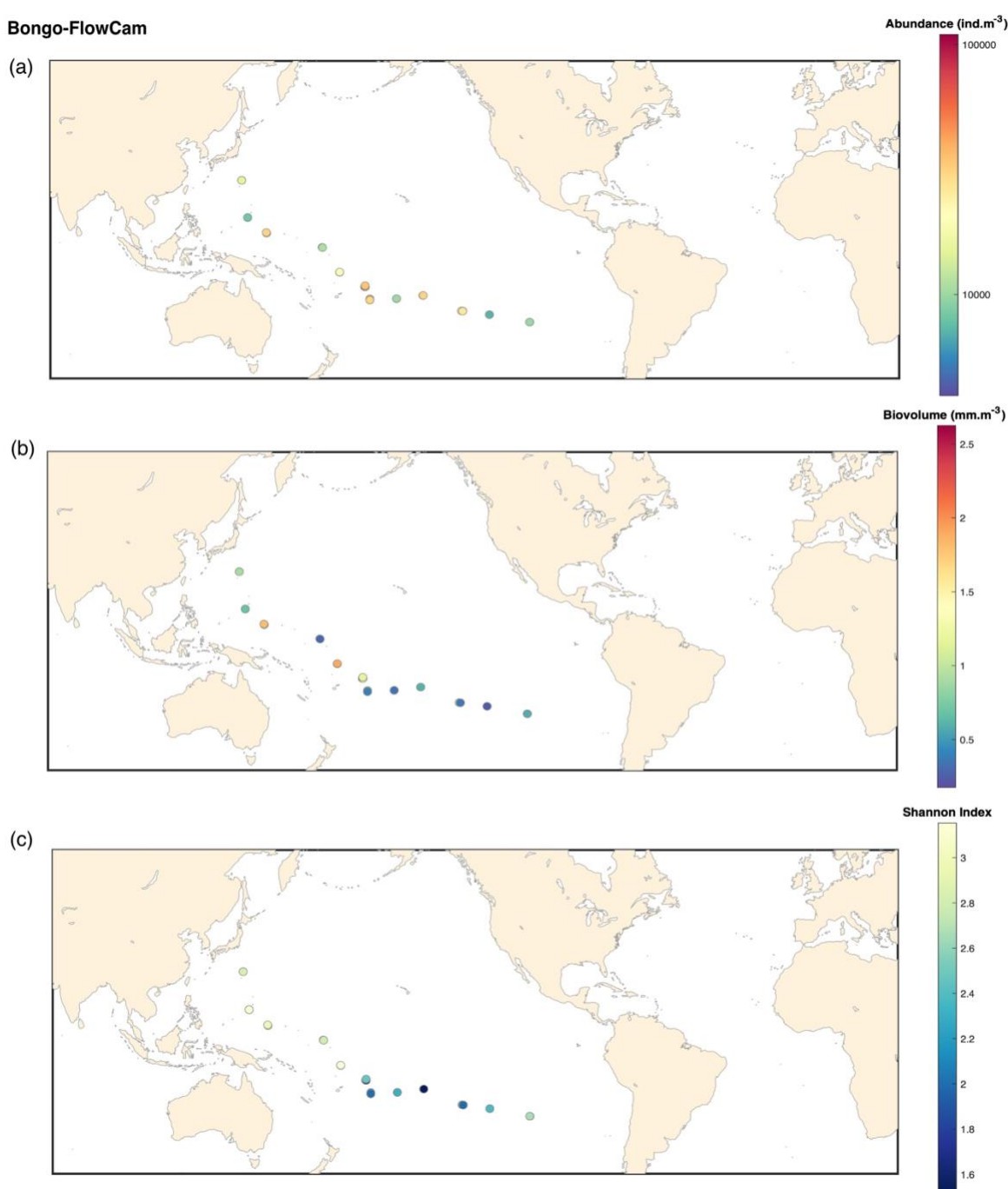

Figure B2. FlowCam Bongo 20 microns: (a) Map of plankton abundance (ind.m$^{-3}$). (b) Map of plankton biovolume
(mm.m$^{-3}$). (c) Map of Shannon diversity Index.

HSN-ZooScan

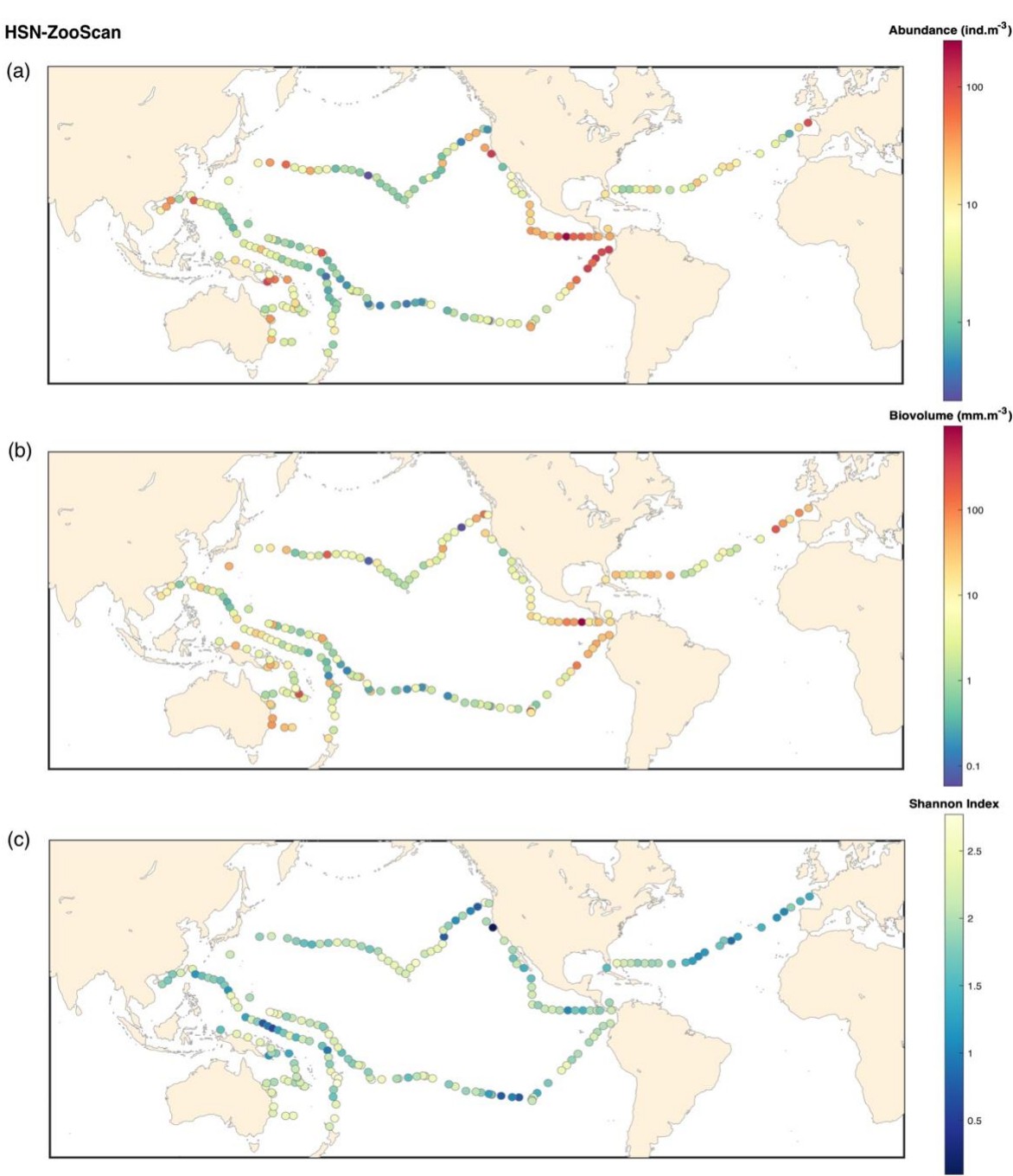

**Figure B3. ZooScan HSN 330 microns: (a) Map of plankton abundance (ind.m$^{-3}$). (b) Map of plankton biovolume**
**(mm.m$^{-3}$). (c) Map of Shannon diversity Index.**

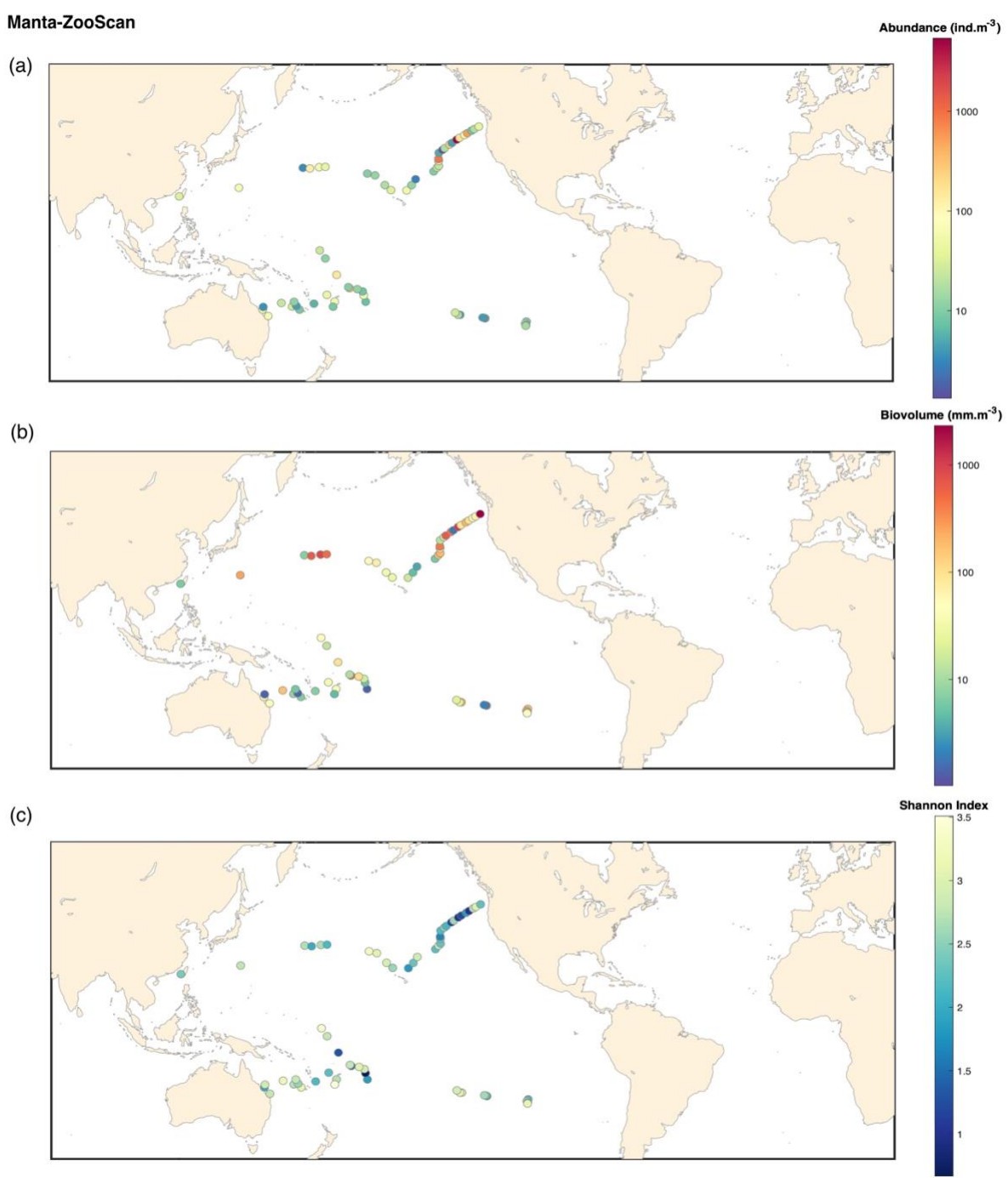

**Figure B4. ZooScan Manta 333 microns: (a) Map of plankton abundance (ind.m$^{-3}$). (b) Map of plankton biovolume**
**(mm.m$^{-3}$). (c) Map of Shannon diversity Index.**

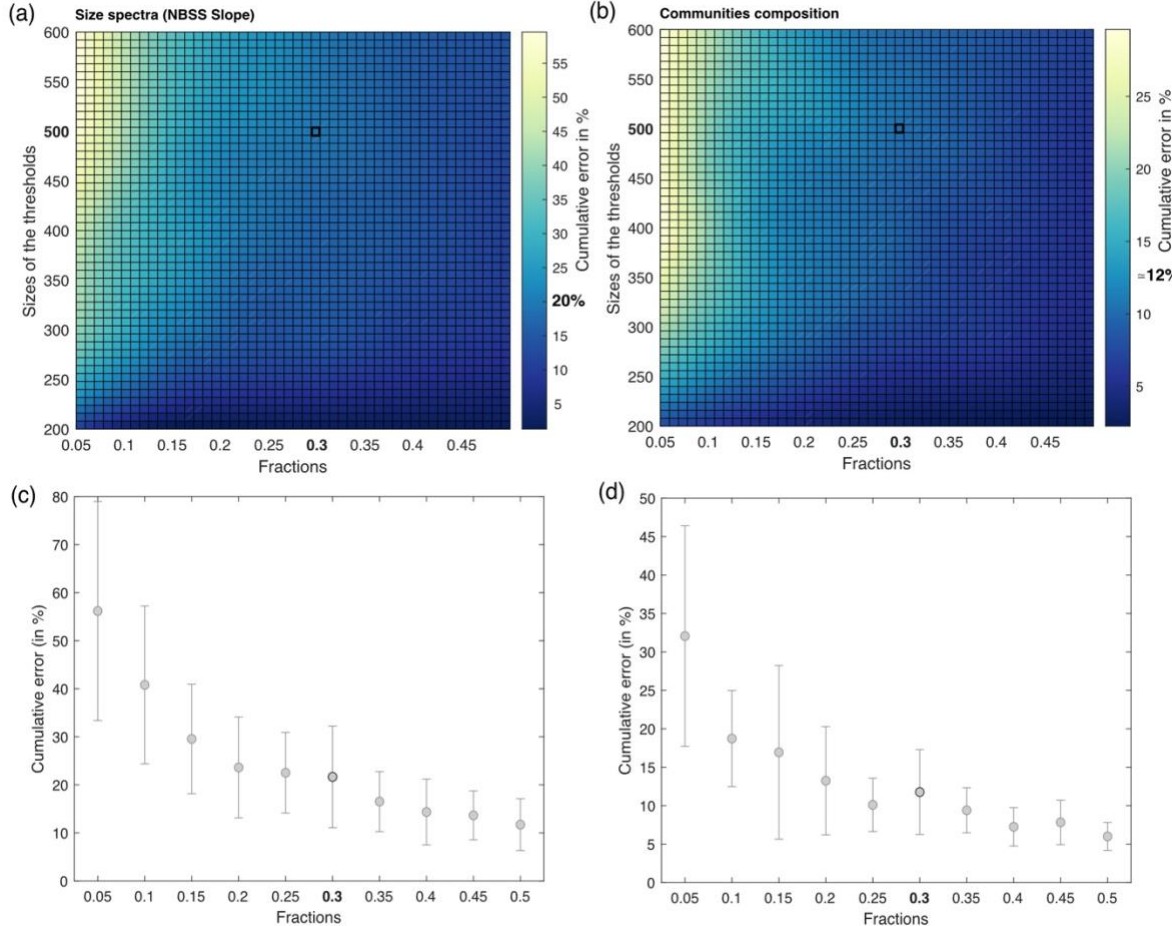

**Figure 4. (a) and (b) Estimated cumulative error associated with partial validation of particles below a size cut-off**
**threshold ranging from 200 to 600 pixels and validated fractions ranging from 5% to 50%. Errors are computed as the**
**percentage Root Mean Squared Error (RMSE) between fully validated samples and partially validated samples in**
**three different metrics for cumulative error in respectively, NBSS slope and communities composition (relative**
**abundance). RMSE values represent the outcomes of simulations, each conducted three times for the four samples,**
**with random sampling. (c) and (d) Cumulative error according to the Fractions chosen in respectively, NBSS slope and**
**communities composition. The threshold is fixed at 500 pixels.**

**Team list**

Tara Pacific Consortium Coordinators:
Sylvain Agostini[5], Denis Allemand[6], Bernard Banaigs[7], Emilie Boissin[7], Emmanuel Boss[3], Chris Bowler[8],
Colomban De Vargas[9], Eric Douville[10], Michel Flores[11], Didier Forcioli[12], Paola Furla[12], Pierre Galand[13], Eric
Gilson[14], Stéphane Pesant[15], Serge Planes[16], Stéphanie Reynaud[17], Matthew B. Sullivan[18], Shinichi Sunagawa[19],
Olivier Thomas[20], Romain Troublé[21], Rebecca Vega Thruber[22], Christian R. Voolstra[23], Patrick Wincker[24], Didier
Zoccola[6]

[5]Shimoda Marine Research Center, University of Tsukuba, 5-10-1, Shimoda, Shizuoka, Japan
[6]Centre Scientifique de Monaco, 8 Quai Antoine Ier, MC-98000, Principality of Monaco
[7]PSL Research University: EPHE-UPVD-CNRS, USR 3278 CRIOBE, Université de Perpignan, France
[8]Institut de Biologie de l'Ecole Normale Supérieure (IBENS), Ecole normale supérieure, CNRS, INSERM,
Université PSL, 75005 Paris, France
[9]Sorbonne Université, CNRS, Station Biologique de Roscoff, AD2M, UMR 7144, ECOMAP 29680 Roscoff,
France
[10]Laboratoire des Sciences du Climat et de l'Environnement, LSCE/IPSL, CEA-CNRS-UVSQ, Université Paris-
Saclay, F-91191 Gif-sur-Yvette, France
[11]Weizmann Institute of Science, Department of Earth and Planetary Sciences, 76100 Rehovot, Israel

[12]Université Côte d'Azur, CNRS, INSERM, IRCAN, Medical School, Nice, France and Department of Medical
Genetics, CHU of Nice, France
[13]Sorbonne Université, CNRS, Laboratoire d'Ecogéochimie des Environnements Benthiques (LECOB),
Observatoire Océanologique de Banyuls, 66650 Banyuls sur mer, France
[14]Université Côte d'Azur, CNRS, Inserm, IRCAN, France
[15]European Molecular Biology Laboratory, European Bioinformatics Institute, Wellcome Genome Campus,
Hinxton, Cambridge CB10 1SD, UK
[16]PSL Research University: EPHE-UPVD-CNRS, USR 3278 CRIOBE, Laboratoire d'Excellence CORAIL,
Université de Perpignan, 52 Avenue Paul Alduy, 66860 Perpignan Cedex, France
[17]Centre Scientifique de Monaco, 8 Quai Antoine Ier, MC-98000, Principality of Monaco
[18]The Ohio State University, Departments of Microbiology and Civil, Environmental and Geodetic Engineering,
Columbus, Ohio, 43210 USA
[19]Department of Biology, Institute of Microbiology and Swiss Institute of Bioinformatics, Vladimir-Prelog-Weg
4, ETH Zürich, CH-8093 Zürich, Switzerland
[20]Marine Biodiscovery Laboratory, School of Chemistry and Ryan Institute, National University of Ireland,
Galway, Ireland
[21]Fondation Tara Océan, Base Tara, 8 rue de Prague, 75 012 Paris, France
[22]Oregon State University, Department of Microbiology, 220 Nash Hall, 97331Corvallis OR USA
[23]Department of Biology, University of Konstanz, 78457 Konstanz, Germany
[24]Génomique Métabolique, Genoscope, Institut François Jacob, CEA, CNRS, Univ Evry, Université Paris-Saclay,
91057 Evry, France

**Author contribution**

Conceptualization and methodology: Tara Pacific Consortium; GB, GG, SA, DA, BB, EB, EB, CB, CDV, ED,
MF, DF, PF, PG, EG, SP, SR, MBS, SS, OT, RT, RVT, CRV, PW, DZ, FL. Samples collection: GB, MLP, AE,
GG. Samples analysis (on lab) and investigation: ZM, NK, LJ, OB, LC, JM, AE. Data analysis, curation and
validation: ZM, NK, GB, LJ, MLP, MP, AE, LKB, FL. Supervision: GG, MLP, LKB, FL. Funding acquisition,
project administration and resources: FL, GG, MLP, LKB, GB, SA, DA, BB, EB, EB, CB, CDV, ED, MF, DF,
PF, PG, EG, SP, SR, MBS, SS, OT, RT, RVT, CRV, PW, DZ. Software: MP, ZM, GB, FL. Visualization and
Writing – original draft preparation: ZM, GB, GG, FL, MP. All authors have read and reviewed the manuscript.

**Competing interest**

The authors declare that they have no conflict of interest.

**Acknowledgment**

Special thanks to the Tara Ocean Foundation, the R/V Tara crew and the Tara Pacific Expedition Participants
(https://doi.org/10.5281/zenodo.3777760). We are keen to thank the commitment of the following institutions for
their financial and scientific support that made this unique Tara Pacific Expedition possible: CNRS, PSL, CSM,
EPHE, Genoscope, CEA, Inserm, Université Côte d'Azur, ANR, agnès b., UNESCO-IOC, the Veolia Foundation,
the Prince Albert II de Monaco Foundation, Région Bretagne, Billerudkorsnas, AmerisourceBergen Company,
Lorient Agglomération, Oceans by Disney, L'Oréal, Biotherm, France Collectivités, Fonds Français pour
l'Environnement Mondial (FFEM), Etienne Bourgois, and the Tara Ocean Foundation teams. Tara Pacific would
not exist without the continuous support of the participating institutes. The authors also particularly thank Serge
Planes, Denis Allemand, and the Tara Pacific consortium. We thank the EMBRC collection CCPv for sample
storage. This work was supported by EMBRC-France, whose French state funds are managed by the ANR within
the Investments of the Future program under reference ANR-10-INBS-02. Support was also provided by the US
National Science Foundation (NSF Biological Oceanography program (grant #2025402 to LKB) and the NASA
Ocean Biology and Biogeochemistry program (grants #80NSSC20K1641). FL  was also funded by the Institut
Universitaire de France and co-funding by the European Union BIOcean5D GA#101059915 and the European

Union's Horizon 2020 research and innovation programme "Atlantic Ecosystems Assessment, Forecasting and Sustainability" (AtlantECO) Grant ID: 862923. FL, OB, ZM are also funded by the ANR grant SmartBiodiv (grant ANR-21-AAFI-0002). This is publication number XX of the Tara Pacific Consortium. The authors particularly thank the Villefranche-sur-Mer Quantitative Imaging Platform (PIQv). Views and opinions expressed are however those of the author(s) only and do not necessarily reflect those of the European Union. Neither the European Union nor the granting authority can be held responsible for them.

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
