# Peer review of "Quantitative imaging datasets of surface micro to mesoplankton communities and microplastic across the Pacific and North Atlantic Ocean from the Tara Pacific Expedition"

_Earth System Science Data, 2024_

## Author Response (AR1)

First of all, the sampling efforts by the TARA expeditions are impressive. I also appreciate a lot the effort put to cover the very wide size spectrum of plankton organisms by the application of several methodological approaches. It is a huge value without any doubt because despite numerous efforts worldwide, a coherent view of the wide size range of plankton realm is still a challenge. The spatial and temporal coverage are impressive as well. This initiative is definitely an important step forward in global plankton research and I am sure it will be used many times at various occasions for a wide range of scientific questions in future as a good reference.

I also value the efforts the authors made to discuss the biases their sampling design could introduce. Especially, the considerations of various effects of flow estimates are a valuable contribution to the surface studies of plankton by different methods.

I have an experience in working with Ecotaxa and their exported files, so I feel quite confident the data quality is good. I checked some of the validation categories of the taxonomical groups I know and they look OK as well.

Thank you very much for your feedback and for recognizing the value of this work. We also hope that the plankton sampling efforts during the Tara Pacific expedition will serve as a valuable reference and be used in the future for a wide range of scientific questions.

We also appreciate your insightful points, which help improve the paper. We have carefully considered each of your points, as detailed below.

**Some minor weak points:**

• Title refers to the Pacific Ocean, but there are also data from the Atlantic Ocean.

Indeed, we do not emphasize the first transatlantic plankton dataset available in our published data. Therefore, we suggest changing the title to: *Quantitative imaging datasets of micro- to meso-plankton communities and surface microplastic across the Pacific and North Atlantic Oceans from the Tara Pacific Expedition.*

• Conclusion is more about the methods than data. To be honest I luck a little bit more statistical summary of the data content (e.g., representation of taxa covered).

We totally agree with your feedback, our conclusion lacks an overall view of the dataset. We have added certain proportions of the data content with a focus on the overall taxonomic composition of the datasets, as well as on the overall trends that emerge from a 1st general overview (see conclusion started lines 614).

• The scientific bias considered is actually only one: underestimation of large fragile gelatinous plankton, but I think it is also important to mention some other problems, such as e.g. time, both time of the day the sampling was performed and time in terms of the season, because the horizontal differences might be also affected by the seasonal

changes and diel migration. The other very important limitation is the fact that the very surface data is not well representative for the zooplankton, which concentrations are also very high within the water column. Of course it does not depreciated the data, but in my opinion should be listed as limitation.

To the first part of your comments: Indeed, the difference in gelatinous fragile organisms is expected and is clear in our results. However, we agree with your feedback that the composition of the plankton between the HSN and Manta nets is more diverse than just the fragile organisms. Our discussion was not clear enough on this point. We have added part of the discussion on a more general difference in commonality than just the fragile organisms. We had a discussion on part 3.2.2 after the description of Figure 7 (lines 564).

Regarding the second part of your comment (surface data are not well representative of zooplankton), thank you for this insightful comment, which is clearly missing from our paper. We have added a comment on this important limitation in section 3.2.2 (end of section; lines 586).

**Some details:**

L48: I am not sure "rich" is the best world... I would suggest 'interesting', or valuable.

**Corrected (line 48)**

L97-98: I think it is crucial to link those data with environmental background data and I would appreciate if the authors could introduce how merging those data can be done – how are they coded to facilitate easy joining etc. Ok, I found such information in methods, but still it could be a little bit more elaborative, explaining very clearly what each number means, is it each station, yes? So, the net sample will get the same number as some measurements? But what about the measurements made constantly?

We completely agree with your comment, thank you. Indeed, the strength of these plankton imaging data is that all the associated environmental data are openly available (https://www.nature.com/articles/s41597-022-01757-w). We suggest adding a short paragraph on data availability to make it easier to establish a link between our plankton imaging data and all the other environmental data available from the expedition (lines 637) However, we refer readers to the paper by Lombard et al. 2023 which describes and references in detail all the environmental (and other) data available from Tara Pacific. More importantly, when inspecting data at the "station" level, all environmental data are already compiled and compatible with an easy analysis and cross analysis (ref to the station level dataset); and when linking with barcodes (an internal reference in all Tara data), it could be further linked to any other connected data (e.g. genomic) by linking it to the sample registery (available within Lombard et al 2023, with sample registery and event registery https://doi.org/10.1594/PANGAEA.944548).

L110: what is high-sampling? You mean high-speed? High-resolution?

Thank you for noticing that. Yes, we were talking about 'high-speed sampling' here. Corrected (line 111)

L124: I would suggest to write 'domains' or 'regimes' instead of 'processes'

Corrected (line 125)

L278: please check this sentence for correctness

Thank you for identifying it. We've corrected the sentence (line 277). Generally speaking, we've reworded the paragraph to be clearer about the description of this quality control.

L280: It is not clear what this quality control step included

We have reworded the paragraph to give a clearer description of this quality control (lines 277 to 284). We hope that this step will be easier to understand.

L340: It is important to remember that Shannon diversity index depends on the number of taxonomical categories and in imaging methods those numbers differ from the typically used taxonomical categories

We agree with your comment. We have added a short comment on the calculation of Shannon diversity in the context of quantitative imaging (lines 362 to 367). Thank you for this comment. We chose the Shannon because this metric is relatively stable and robust to the sampling effort (both in term of number of organisms counted, but in some respect also to the taxonomic depth reached). Although it should be noted that most of the time the same may apply to genomic samples where OTU is not fully linked to the species, but in general we achieve a relatively good correspondence in between the different diversity (but see Ibarbatlz et al 2019, and in particular the figure S3 and S6H.

L380: maybe despite clogging the difference is in depths investigated? (deeper layers with divers compared to the shallow pump system)

The point about the depth of the relationship between DN and Bongo has rightly been raised. However, the difference in investigated depths does not seem significant between the two sampling methods. The Deck-Net pump collected water from the first meter of the water column, while the diver-operated Bongo net sampled at approximately two meters below the sea surface, resulting in a potential difference of only 1-2 meters at most.

For us, a notable difference between these two nets is the location of their deployments, which correspond to distinctly different environments: Bongo nets near islands, reefs, or

within lagoons, whereas the Deck Net was deployed in the open ocean. We have developed this idea and added references (lines 386 to 397) to our two new parallel studies in progress, one of which is available as a pre-publication (Bourdin et al., 2024) and the other is forthcoming (Kristan et al.), which highlight the higher concentration of plankton in the Bongo sampling zones, as opposed to the Deck-Net zones, which would be responsible for Bongo clogging.

L411: 'to optimise'

**Corrected (line 419)**

Fig. 7. Knowing that both nets, HSN and Manta, had the same mesh size, but just different speed and opening area, this difference in plankton composition is larger than expected and does not concert only the gelatinous forms

See our previous answers in the "minor weak points" part, we fully agree with your comment and add a more detailed discussion on this point.

L557: 'serve' instead of 'serves'

**Corrected (line 624)**

L558: 'serve' instead of 'serves'

Corrected (line 625)

**General comments**

This paper presents combined, basin scale wide, datasets of surface zooplankton and phytoplankton, obtained using imaging instruments and associated computer assisted curation techniques. It also features a microplastic dataset of, however, less important spatio-temporal coverage. The originality of these datasets resides in their wide coverage of the Pacific Ocean and their novel focus on the pacific islands. The combined phytoplankton and zooplankton datasets are rich and spatially complex as they feature data originating from multiple and non-standard sampling instruments and multiple environments (open ocean, upwelling systems, and islands and lagoons), and participate in making plankton data from the under sampled Pacific Ocean available to the community. This paper also presents a wide range of methods, from sampling to curation and analysis methods for large and complex imaging dataset.

\*As such, the data presented in this paper, and the paper itself, are a substantial contribution to the enhancement of the representation and availability of planktonic data for the global marine science community\*.

\**The presented data is accessible through the links provided in the data availability section, and is consistent with what is presented in the paper*\*

However I do have a concern about the "quantitative" nature of the data presented in the paper and associated datasets, particularly for the zooplankton datasets. The paper and associated datasets provide all the necessary information to calculate quantitative estimates of the samples contents, indeed. Yet, I question the quantitative nature of the sampling itself. Phytoplankton is known to be heterogeneously distributed from the surface to deep waters, especially in transparent waters such as those of the pacific gyres, where it shows marked Deep Chlorophyll Maxima sometimes tens to hundred meters below the water surface; zooplankton exhibits large diel vertical migrations, especially in transparent waters, where it can be aggregated several hundred of meters below the surface during daytime. It follows that, although being "quantitative" (enabling the calculation of estimates), are the datasets "representative"? I suggest to formulate a clear statement in the introduction confronting the two notions (quantitative vs representative) and to elaborate a paragraph in the discussion to highlight the possible limitations of the use of the presented data in various contexts, i.e. biogeochemistry, modelling or biodiversity studies, etc.

\*Besides this semantic concern, I consider the data presented here of great value, the paper sound and clear, and suitable for publication after a revision addressing the above mentioned comments and the specific and technical comments below\*.

**Dear Reviewer 2,**

Thank you very much for recognizing the contribution of our paper, as well as your positive comments on our paper.

We greatly appreciate all the specific comments you provided section by section, which have been very beneficial to our study. These comments have allowed us to deepen our discussion on the advantages and limitations of our dataset, really improving the paper. We have followed your recommendations point by point, including restructuring the introduction to focus on surface and daytime plankton sampling, thus addressing the limits of "representativeness" of our dataset over the whole planktonic ecosystem. In more general, we have, following your suggestions, added a new section titled "4. General Discussion" to cover all the important points to be discussed around our datasets, including the question of representativeness, as well as opening up the use of our data in various studies.

We hope this revision meets all of your expectations.

Sincerely,

Specific comments

**Titles and Abstract**

The paper title should mention that the data presented in the paper is about surface plankton only, as it can be misleading for readers. Net collected surface plankton datasets are not standard, regarding the other datasets available in the literature.

We totally agree that the focus on surface plankton sampling was not explicitly reflected in our title and should be clarified. Therefore, we propose adding "surface" to the title. Additionally, following the recommendation of Reviewer 1, we also include "Atlantic" to better represent the dataset's geographic coverage.

The revised title is:

"Quantitative imaging datasets of surface micro- to mesoplankton communities and microplastic across the Pacific and North Atlantic Ocean from the Tara Pacific Expedition"

Also, the datasets are all untitled as "Global", which is not the case. They cover the Pacific Ocean, pacific islands and a North Atlantic transect. I suggest replacing "Global scale" in all the datasets titles by "Ocean wide" to better fit to the real data coverage.

We also agree with your suggestion regarding dataset titles. To better reflect the actual coverage, we are following your recommendation to replace "Global scale" with "Ocean-wide." We have initiated the request to update the dataset titles on the Seanoe platform.

For example, the revised title for the HSN-ZooScan dataset (doi 10.17882/102336) would be:

"Ocean-wide surface mesoplankton dataset collected with a High-Speed Net and imaged with ZooScan during the Tara Pacific Expedition"

L 21-23: The abstract should mention that the dataset also covers a North Atlantic transect.

Indeed, we have updated the abstract to include the mention of the "North Atlantic" transect.

*.L* 33: There should be a mention here that a fraction only of all the imaged objects' taxonomic identifications were visually validated or corrected by experts (phytoplankton).

We have added a clarification stating that for all microplankton (FlowCam images) smaller than 45  $\mu$ m, only a subsample of 30% of the annotations was 100% visually validated by experts.

For clarification, all samples for the Deck-Net FlowCam were validated taxonomically (with this nuance on organisms smaller than 45  $\mu$ m: 30% visually validated then extrapolated). For the Bongo samples, there are still samples that have not been validated (8 samples) and therefore not published. The samples validated by the Bongo-FlowCam also had this nuance: on organisms smaller than 45  $\mu$ m: 30% visually validated then extrapolated. As the chosen platform is updatable, we will add these new samples when they are validated.

L 39-41: I suggest to move this sentence above in the abstract (L 31, after ZooScan system?)

We have moved the sentence to line 30, as suggested.

**Introduction**

The introduction is already clear and sound, yet a bit long. I suggest a shortening by moving large parts of the L85 to 109 to the discussion and method sections. Most of this paragraph is already repeated in the method & discussion. On the contrary, I suggest a short elaboration on the sampling biases (L 109-112) to formulate a reflexion on the interplay between the datasets spatial (vertical & horizontal) resolutions, and representativeness of the datasets in relation to current scientific questions and needs.

We agree with these comments in general and appreciate this insightful feedback.

Therefore:

 We have removed the section (lines 85-109) that describes the datasets, as it was indeed repeated in other parts of the manuscript. We ensured that all additional information was included in the methods and/or discussion sections and added anything that was missing.
Thank you for this insightful comment. We fully agree with taking it into account in our study. We propose to introduce the various limitations associated with these datasets in the final paragraph of the introduction (limitations of high-speed sampling and the representativeness of daytime, surface plankton), and then to discuss these limitations in more detail in the discussion section, as Reviewer 1 had also pointed out.

*L* 55: which fisheries, and where? please add a couple references.

We agree with this suggestion and have added a more explanatory sentence on the importance of plankton in fisheries management, along with relevant references (lines 56-57).

Methods

*L122-125: repetition from what is in the introduction.*

Also, the sampling description needs a bit more elaboration to describe the sampling strategies: why daily frequency? during daytime or night? etc.

We agree with these comments. To address the repetitions (L122-125), we have removed them in the introduction. In addition, we have added the required sampling description (daily frequency and time of day).

Figure 2: I suggest a reorganization of panels to show sequentially current panels a & c (continuous sampling), followed by b & d (more discrete sampling, Islands). a & c show the same sampling strategy while b & d show a similar sampling strategy yet different from what is shown in a & c.

The reorganization has been made in the figure and the associated legend to reflect the sequence as you suggested: panels a & c (continuous sampling) are now followed by panels b & d (more discrete sampling, islands).

Figure 3: Amazing figure! Lots of information. However, I suggest to remove any written mention of protocols not explained in this paper (i.e. [S300], etc.). Also: panel 2c: [H20] instead of [H2O]

Thank you for your comment on this figure! We have removed the mention of protocols not explained in this paper (i.e. [S300], etc.) and corrected the notation in panel 2c to use [H20] instead of [H2O]. Thank you for pointing that out.

L 199-202: Please consider writing equations with terms, terms definitions, and numerical values when necessary instead of numerical values and operational variables names. What does "0.3" represent in equation 2,3 and 4?

We have modified the equations to include operational variable names instead of numerical values. Regarding the coefficient "0.3" in equations 2, 3, and 4, it represents the impeller pitch recommended for adjusting flowmeter towing distance estimations by Hydrobios. The flowmeters used during Tara Pacific were from Hydrobios, and according to their manual (available here), the impeller advances 0.3 meters per revolution. Therefore, the number of revolutions multiplied by 0.3 gives the towing distance. We have added a sentence to clarify this in the manuscript.

L 279: What does this quality control consist of?

We have provided more details on this quality control process in the manuscript, as we agree that further clarification was necessary (see also our answers to the reviewer 1).

L 291-292: "...but the list ... common". Please move to L 287-289.

Corrected.

L 296-319: This whole paragraph deserves a specific section (2.4.1 or 2.5)

We agree with this suggestion and have created a specific section for this paragraph. It has now been moved to Section 2.5, titled 'Case study of FlowCam taxonomic identification for objects smaller than  $45 \mu m'$ .

L 316-319: 20% error on which NBSS parameter? 12% error on relative abundance on which variable(s)? A reproduction of figure 4 a & b for NBSS parameters would be interesting to see (in the appendices?). I let the author choose to show it or not.

Indeed, this needed clarification. When comparing size spectra, we specifically refer to the slope of the NBSS (corrected in the manuscrit). We proposed to add this analysis in appendix C, a supplementary figure that shows reproduction of panels a and b for NBSS slope, and for relative abundance also.

L 332: First sentence is a repetition. Please remove from MS.

Corrected, thanks for pointing it out.

*L* 332-337: Please provide a link to the Ecotaxa projects in the section data availability, for each dataset.

We have added the links in the data availability section for each dataset.

*L* 346: *Please provide the ESD calculation methods used.*

We have added a new line in Table 1 (Formulas used to calculate quantitative variables in datasets) detailing the ESD calculation method along with the definition of the parameters used.

*L* 353: Please provide a couple of references on Elton's pyramids and NBSS theoretical and/or practical links.

We have added references on Elton's pyramids and the theoretical and practical links of NBSS to support this section.

Technical validation and discussion

L 363: ... sampling for \*phytoplankton\* quantitative...

Corrected.

L 366-367: ... the transects \*sampled\* through the net stations?...

*L* 372-373: similar trends between what and what? Please add details and reformulate this whole sentence as it lack clarity.

*L* 373-374: Overestimation of the volume sampled by the Bongo net compared to that of the Deck-Net. Am I right? Please reformulate for more clarity.

For these three points, and in alignment with the comments from Reviewer 1, we have revised the paragraph in the manuscript to improve clarity and enhance readability.

*L* 374: The authors should mentioned the divers earlier in the manuscript, in the sampling section.

Indeed, we add 'towed by divers using underwater scooters' on the Methods section (2.1.2 Bongo nets sampling).

*L* 377-383: Not clear enough. What are the length and filtration area of the nets? What is the calculated filtration efficiency ratio of the used bongo net? is it close from that of Smith et al, 1968 (~1380)?

Thank you for your relevant comment to improve this study. In response to your point, we realized that calculating the filtration efficiency was not reliable in our case. Smith et al. (1968) defines filtration efficiency as the ratio of the theoretical filtered volume to the encountered volume (measured by flowmeters)  $\times$  100. However, in our case, this calculation is not applicable because we lack confidence in both of these volume estimates:

- Flowmeter limitations: The Bongo nets were equipped with flowmeters rated for speeds above 0.3  $m \cdot s^{-1}$ , but the relatively low towing speed of the underwater scooter was insufficient to generate enough water flow through the 20  $\mu$ m mesh to rotate the flowmeters reliably.
- Uncertainty in theoretical volume: The deployment time of the Bongo nets by divers was highly uncertain. This uncertainty arose from inconsistencies in how divers recorded deployment duration and from methodological biases linked to using an underwater scooter, making the estimated filtered volume unreliable. Moreover, the suspended particle concentrations were very variable for different sampling sites which complicated the correct prediction of the towing time required to obtain reasonable concentration in the net and avoid clogging.

Overall, the lack of correlation of total chlorophyll a and total phytoplankton biovolume from FlowCam (as shown in Figure 5) shows that the Bongo net sampling was not quantitative. The correlation between chlorophyll a (x-axis) and total phytoplankton biovolume (y-axis) of the Bongo being lower than the Deck-Net samples on the y-axis, phytoplankton biovolume was underestimated relative to chlorophyll a in the Bongo samples compared to Deck-Net samples, which we know is more quantitative by design. Considering the methodological limitations of the Bongo net filtration volume estimation, the most reasonable explanation for this is an overestimation of the theoretical volume that can be due to clogging.

We have updated and improved the whole discussion paragraph on this issue in the manuscript.

Figure 5. panel a: the y-axis unit could be simplified as m-3. What does "equivalent abundance" means? I am not sure it is relevant here. Panel b: the y-axis unit is not a biovolume unit, it is an abundance unit, please correct the label or the unit, and the text and the legend accordingly. Panels a & b: how many samples are represented for each device?

We have simplified the y-axis unit to m-3 and removed 'equivalent abundance' (thus, also corrected in the figure 7a). We initially used this term because the NBSS unit is mm3.mm-3.mm-3, which, as you pointed out, is equivalent to xx.m-3 and thus analogous to abundance in ind.m-3.

Additionally, we have corrected the y-axis unit in panel b (biovolume in mm3.m-3) to ensure consistency in the text and legend. Lastly, we have specified the number of samples represented for each device in the legend: "Bongo nets (34 samples) and the Deck Net (207 samples)".

*L* 389-390: Does that mean that the data presented in this figure originate from the 4 samples 100% taxonomically validated?

These data represent all the phytoplankton samples from the Bongo and Deck-Net samples. Both types of samples were analyzed by the FlowCam, meaning they were split into images < 45 microns and > 45 microns. For the > 45 microns images, 100% were taxonomically validated, while for the

Following your recommendation of plotting the "instrumental individual size distributions," with plotted median per size class for all HSN and Manta samples, we observe the quantitative difference seen in the intercepts as well as the overlap in size spectra. We also noticed a more pronounced difference for larger size classes, where the HSN lacks data points compared to the Manta, which is indicative of the HSN not sampling larger and more fragile organisms.

Thus, we have modified the manuscript to make the differences in NBSS parameters clearer and more precise.

*Figure 7: what does size (\mum) stands for in the x-axis label? ESD, body length?*

Size in ESD ( $\mu m$ ), we have updated the figure 7 to make it clearer.

*L* 526-538: a map showing the abundances at each sampling point would be appreciated to accompany this paragraph

We refer readers to the maps of abundances, biovolumes, and diversity at each sampling point for each dataset (DN, HSN, Bongo, and Manta) presented in Appendix B, which includes 13 descriptive maps. We have clarified this in the manuscript to make it more explicit.

Overall, this section is technically convincing, but the discussion lacks a paragraph addressing the representativeness and usefulness of those datasets in various scientific context (see general comment). Finally, it would be much appreciated to show a figure combining phyto-and zooplankton data at the Pacific Ocean scale. Showing such a figure would be truly novel: combined, in-situ, simultaneous, observation of phyto- to zooplankton based on physical sampling and imaging at a basin scale is unprecedented.

**Conclusion**

L 540-542: I would reformulate this sentence in a more modest manner. Sexy sentence, but out of the scope of the paper and the data presented in the paper.

Indeed, we have modified the sentence to: "*The Tara Pacific Expedition is part of the first initiatives aiming to implement a system for discrete sampling of the planktonic ecosystem while operating at cruising speed (5–9 knots), covering viruses to metazoa at the scale of the whole expedition (Gorsky et al., 2019) and focusing on micro- to mesoplankton in this paper.*" This revision clarifies the distinction between the broader expedition objectives and the specific scope of this study.

**Data availability**

Please provide the links to access the Ecotaxa projects associated with the presented datasets.

**Corrected.**

Please also check and add the units (if necessary) of the quantitative variables presented in the data files, in the data files headers, and specify if the sampling hour are local time or UTC, within the data files headers as well.

We have reviewed all units of quantitative variables in the data file headers. Regarding the sampling hours, they are in UTC, but this was not clearly indicated in the published files. Thank you for pointing this out. We will add this clarification to the data files. Additionally, for the latitude and longitude coordinates, we will also specify that the unit is in decimal degrees (DD). We have initiated the request to update these units and dataset titles (as mentioned above) on the Seanoe platform.

**Technical comments (mostly suggestions)**

Thank you for these technical comments point by point. We have taken all of them into account in the revised manuscript.

- L 31: ... ZooScan system, back on land.
- L 45: replace trophic chain by food webs
- L 46: replace socio-ecosystemic interests by ecosystem services
- L 47: replace propose by present ; cover ; from 20 µm to ... at the scale of...
- L 110: high-resolution or high-speed sampling
- Figure 2: replace traject by route
- L 155, L 179: starboard side of the ship

- L 248: replace has a tendency to crystallise by may form crystals grains
- L 249: be digitised
- L 250: Thus they may represent a large ...
- L 276-278: touching or overlapping objects?
- L 276: Replace affect by bias
- L 278: ZooProcess digital separation ...
- L 290: the published SEANOE dataset will be updated accordingly
- L 354: ...provided in a separate zip file. Remove Tables (.csv)
- L 365: The bongo was deployed on reefs or in lagoons
- L 366: chlorophyll a
- L 380: filtering area
- *L* 453: a clear discrepancy
- L 465: ... generates turbulence and could affect...
- L 487: ...studying surface plankton and widely recognised...
- L 507: ... the slopes of the HSN's equivalent abundance are steeper than ...